# Arctic sea ice mass balance in a new coupled ice-ocean model using a brittle rheology framework

Guillaume Boutin[1], Einar Ólason[1], Pierre Rampal[2], Heather Regan[1], Camille Lique[3], Claude Talandier[3], Laurent Brodeau[2], and Robert Ricker[4]

[1]Nansen Environmental and Remote Sensing Center, and the Bjerknes Center for Climate Research, Bergen, Norway
[2]CNRS, Institut de Géophysique de l'Environnement, Grenoble, France
[3]Univ. Brest, CNRS, IRD, Ifremer, Laboratoire d'Océanographie Physique et Spatiale, IUEM, Brest 29280, France
[4]NORCE Norwegian Research Centre, Tromsø, Norway

**Correspondence:** Guillaume Boutin (guillaume.boutin@nersc.no)

**Abstract.**

Sea ice is a key component of the Earth's climate system as it modulates the energy exchanges and associated feedback processes at the air-sea interface in polar regions. These exchanges have been suggested to strongly depend on openings in the sea ice cover, which are associated with fine-scale sea ice deformations, but the importance of these processes remains poorly understood as most numerical models struggle to represent these deformations without using very costly horizontal resolutions ($\simeq$5 km). In this study, we present results from a 12 km resolution ocean–sea-ice coupled model, the first that uses a brittle rheology to represent the mechanical behaviour of sea ice. This rheology has been shown to reproduce observed characteristics and complexity of fine-scale sea ice deformations at relatively coarse resolutions. We evaluate and discuss the Arctic sea ice mass balance of this coupled model for the period 2000–2018. We first assess sea ice quantities relevant for climate (volume, extent and drift) and find that they are consistent with satellite observations. We evaluate components of the mass balance for which observations are available, i.e. sea ice volume export through Fram Strait and winter mass balance in the Arctic marginal seas for the period 2003–2018. Model values show a good match with observations, remaining within the estimated uncertainty, and the interannual variability of the dynamic contribution to the winter mass balance is generally well captured. We discuss the relative contributions of dynamics and thermodynamics to the sea ice mass balance in the Arctic Basin for 2000–2018. Using the ability of the model to represent divergence motions at different scales, we investigate the role of leads and polynyas in ice production. We suggest a way to estimate the contribution of leads and polynyas to ice growth in winter, and we estimate this contribution to add up to 25%–35% of the total ice growth in pack ice from January to March. This contribution shows a significant increase over 2000–2018. This coupled framework opens up new opportunities to understand and quantify the interplay between small-scale sea ice dynamics and ocean properties.

## 1   Introduction

Arctic sea ice is a key component of the global climate system that has been undergoing rapid changes during recent decades (IPCC, Meredith et al., 2019). Its evolution is driven both by thermodynamics (ice growth and melt) and dynamics (ice frac-

turing, divergence and convergence). At small scales and in the ice pack, sea ice dynamics are characterised by the occurrence of fractures and the formation of ridges and leads, resulting in velocity discontinuities usually referred to as Linear Kinematic Features (LKFs, Kwok et al., 1998). These ubiquitous features, particularly leads, are expected to have a strong impact on ocean-ice-atmosphere interactions in polar regions (Lüpkes et al., 2008; Marcq and Weiss, 2012; Steiner et al., 2013)), but this impact at a pan-Arctic scale has not yet been quantified. To assess whether this impact is significant or not, numerical models need to represent the heterogeneity associated with LKFs in the ice cover, and therefore ensure a correct simulation of small-scale ice dynamics.

The reproduction of the observed sea ice cover heterogeneity in models remains a challenge (Blockley et al., 2020; Hunke et al., 2020; Hutter et al., 2022), unless they use horizontal resolutions higher than $\simeq 5$ km (Bouchat et al., 2022; Hutter et al., 2022). Using such a high-resolution grid is very costly and therefore not always suitable for simulations over long periods and/or large domains. LKFs are related to the mechanical behaviour of the sea ice, and their under-representation in models may therefore be linked to a misrepresentation of this behaviour by the rheologies used in these models (Girard et al., 2009), which are generally using a visco-plastic (VP) framework (Hibler III, 1979). Recent efforts have focused on finding the best way to represent LKFs within a VP rheology framework (e.g. Mehlmann et al., 2021; Ringeisen et al., 2021). An alternative approach would be to use a brittle rheology framework, which has been shown to successfully reproduce LKFs at relatively coarse resolutions (for instance 10 km in Rampal et al., 2019).

Brittle rheologies are relatively new in sea ice modelling, and significant progress has recently been made (Girard et al., 2011; Bouillon and Rampal, 2015; Dansereau et al., 2016; Ólason et al., 2022; Plante and Tremblay, 2021), enabling their use in pan-Arctic process studies (Rampal et al., 2016, 2019; Ólason et al., 2021; Rheinlænder et al., 2022). However, most of these studies have focused on sea ice deformations and periods of time equal to or shorter than a year. Using a stand-alone version of the *neXt generation Sea Ice Model* (neXtSIM) with the Brittle-Bingham-Maxwell (BBM) rheology, Ólason et al. (2022) were able to reproduce the evolution of Arctic sea ice volume over two decades as well as important statistical characteristics of sea ice deformations. However, the impact of using such a rheology in a large-scale model on the Arctic sea ice mass balance has not yet been assessed.

Choosing which rheology to use in a sea ice model is likely to have an impact on the modelled sea ice mass balance in the Arctic. One of the reasons is that the internal stress of the ice, the term related to the sea ice rheology in the momentum equation, impacts the net transport of ice between regions (Steele et al., 1997). This net transport has an impact in the regional mass balance that can be comparable to the thermodynamics (Ricker et al., 2021). The importance of the internal stress in controlling the large scale pattern of ice thickness and in the modulation of the ice export through Fram Strait has also been shown in a study by Spall (2019), using scale analysis and an idealized model. Spall (2019) also stresses the close interplay between sea ice dynamics and thermodynamics at large scales.

Fine-scale sea-ice dynamics also impact the sea ice mass balance. Divergent features in the ice, like leads and polynyas, are associated with localised intense ocean heat loss that enhances sea ice production in winter (Kwok, 2006; Wilchinsky et al., 2015; von Albedyll et al., 2022). The magnitude of ice production in leads remains largely uncertain. Kwok (2006) have estimated this contribution to $\simeq$30% of the total ice production in pack ice from November to April in the western part of the

Arctic Basin for the period 1997–2000. More recently, von Albedyll et al. (2022) also estimated this contribution to be around 30% during the Multidisciplinary drifting Observatory for the Study of Arctic Climate (MOSAiC) campaign. These estimates suggest that properly representing ice formation in leads is key to ensuring a realistic magnitude and distribution of ice growth in numerical models. In return, sea ice models that are able to capture such features can assess the importance of leads for ice formation at large scales and over long periods of time, complementing observations when they are missing.

In this study, we investigate the Arctic sea ice mass balance from 2000 to 2018 in a new ocean–sea ice coupled system which uses the ocean component of the Nucleus for European Modelling of the Ocean (NEMO) system and the latest version of the neXtSIM (v2, Ólason et al., 2022). This is the first ocean–sea-ice coupled platform that includes a sea ice model with a brittle rheology. The main objective of this study is to use a coupled ice-ocean modelling system to examine the ice mass balance in the Arctic over the period 2000–2018, and assess the relative importance of the different source and sink terms of that balance in this rheological framework. After a description of the modelling setup, we evaluate the simulated sea ice volume, extent and drift against available observations, as well as the dynamic and thermodynamic contributions to the winter ice mass balance estimated by Ricker et al. (2021) and the sea ice transport through Fram Strait. We then discuss the Arctic mass balance for the whole study period, with a focus on the impact of openings associated with small-scale dynamics (leads and polynyas) in winter.

## 2    Description of the new coupled sea ice—ocean model

### 2.1    Model components

The ocean model is OPA, which is part of the NEMO3.6 modelling platform (Madec, 2008). We use the regional CREG025 configuration (Talandier and Lique, 2021), which is a regional extraction of the global ORCA025 configuration developed by the Drakkar consortium (Barnier et al., 2006). It encompasses the Arctic and parts of the North Atlantic down to $27^o$, and has 75 vertical levels and a nominal horizontal resolution of $1/4^o (\simeq 12$ km in the Arctic basin). Initial conditions for the ocean are taken from the World Ocean Atlas 2009 climatology for temperature and salinity. For the lateral open boundaries, monthly climatological conditions (comprising sea surface height, 3-D velocities, temperature and salinity) are taken from a long ORCA025 simulation performed by the Drakkar Group.

The sea ice model is the version 2 of the neXtSIM model as recently presented in Ólason et al. (2022). It uses the Brittle-Bingham-Maxwell (BBM) rheology to represent sea ice dynamics. Sea ice thermodynamics are the same as described in appendix A2 of Rampal et al. (2019). In short, the model considers three ice categories: "thick" ice, open water, and newly formed "young" ice. The young ice is made of ice formed from the super-cooling of open water: this ice category is associated with Marginal Ice Zones (MIZs) and openings in the ice cover (leads and polynyas). This scheme is used to represent the rapid growth of newly formed ice (young ice, frazil, nylas, etc.), from a minimum thickness $h_{\min}$, set to 5cm, to a maximum thickness $h_{\max}$, that corresponds to the transition to thicker, consolidated ice. Young ice is redistributed in the "thick" ice category once its thickness exceeds $h_{\max}$. Increasing $h_{\max}$ enhances ice growth in the winter. We found that a value of $h_{\max}$=18 cm gives a

reasonable winter sea ice thickness in our simulation (see section 4.1). The source and sink terms from the thermodynamics are computed by applying the zero-layer Semtner (1976) vertical thermodynamics to the young ice category and that of Winton (2000) for the thick ice. We do not use an explicit melt-pond scheme, but the albedo scheme we use (the same as the standard albedo scheme "ccsm3" used in CICE, Hunke et al., 2017) accounts for the effect of melt ponds by reducing the albedo value when the surface temperature of sea ice increases. It is likely that the use of an explicit melt-pond scheme (e.g. Flocco et al., 2010), or more complex representations of processes related to brine (Vancoppenolle et al., 2009) (instead of a constant salinity here) or snow would affect the sea ice extent and thickness in our results, but the effect of using another parameterization could only be assessed after a re-tuning of the model (as in Zampieri et al., 2021).

The main (advection) model time step is 450s, with 120 sub-cycles used to solve the dynamics resulting in a dynamical time step of 6s. The use of a coupled system has required some changes to the default values of the sea ice parameters that were used for simulations using neXtSIM in its stand-alone configuration (as in Rampal et al., 2019; Ólason et al., 2022). These changes are summarised in Table 1. For the dynamics, our setup is very similar to the one described in Ólason et al. (2022), with the exception of small changes in the values of the ice-ocean and ice-atmosphere drag coefficients and a decrease of the value of the scaling parameter for the ridging threshold, $P$. These changes are needed to ensure that the ice–ocean and ice–atmosphere stresses are properly balanced against the internal ice strength, since this balance is different when neXtSIM is coupled compared to a stand-alone setup. The stress values are chosen to match the observed large scale drift and thickness distribution as well as possible, while still maintaining good deformation patterns and statistics (see Appendix A). We use the ice grounding scheme from Lemieux et al. (2015) to represent landfast ice (as in Rampal et al., 2016), with a critical thickness parameter for ice grounding of $k_1 = 5$ (see Lemieux et al., 2015). The initial fields of sea ice thickness and concentration are taken from the same long ORCA025 simulation as used for the ocean lateral boundaries and climatological conditions.

## 2.2 Coupling between the Lagrangian sea ice and the Eulerian ocean models

OPA and neXtSIM are coupled via the OASIS-MCT coupler (Craig et al., 2017). The coupling time step is taken to be equal to the ocean model time step (twice the ice model time step) of 900 s. As summarised in Figure 1, OPA receives all the required information about surface fluxes (heat and salt) and stresses from neXtSIM. In return, OPA sends back information about properties of the ocean surface to neXtSIM. This includes sea surface temperature, salinity, height, and currents, as well as the absorbed fraction of net solar radiation. NEMO includes a coupling interface to run the component model OPA and an ice model (LIM3, SI[3], or CICE) coupled through OASIS. We make use of this coupling interface here, albeit with the minor modifications required to allow for vector orientation on the ocean and ice model grids to be different (as is implicitly assumed in the NEMO code).

One of neXtSIM's particularities is that it uses a Lagrangian moving mesh (Rampal et al., 2016). This ensures little numerical diffusion, which is a condition required to obtain a good localisation of sea ice deformations. However, this particularity makes the interface between neXtSIM and OASIS more complex than a standard coupling interface, as OASIS is not able to accommodate a moving mesh. Therefore, we chose to implement a fixed exchange grid within neXtSIM, which we use to interface with OASIS. For this exchange, neXtSIM interpolates all coupling quantities between the moving mesh and the exchange

grid internally, while all communications with OASIS are carried out on the exchange grid (Figure 1). This interpolation is done by averaging exchanged quantities weighted by the area of overlap between triangles of the mesh and quadrangles of the exchange grid, in a conserving way. The interpolation weights are recomputed after each Lagrangian regridding.

Heat fluxes between the ocean, ice, and atmosphere are computed using traditional bulk formulae. For ocean-atmosphere exchanges, bulk formulae from OPA have been implemented in neXtSIM. This was done using the AeroBulk library (Brodeau et al., 2017), on which OPA has relied since version 4. The bulk formulae for atmosphere-ice fluxes are described in Rampal et al. (2016, 2019). The bulk formulae for ice-ocean fluxes are the same as in the Louvain-La-Neuve sea Ice Model (LIM) version 3.6 (Rousset et al., 2015).

### 2.3   The Arctic simulation and regions studied

The model simulation, hereafter referred to as OPA-nex, starts on 1st January 1995 and runs until 31st December 2018. Atmospheric forcings are taken from the hourly, 1/4 degree horizontal resolution, ERA5 reanalysis (Hersbach et al., 2020). We exclude the first 5 years of the simulation from the analysis to account for model spin-up. To estimate the length of the spin-up, we applied different initial ice conditions on 1st January 1995, and found no sensitivity of our results over the period 2000–2018 . All output variables from neXtSIM are interpolated using a conservative scheme from the moving Lagrangian model mesh onto a fixed and regular Eulerian grid and are averaged on a 6-hourly basis.

Our analysis focuses on sea ice properties in the Arctic region (Figure 2) where sea ice deformations in neXtSIM have been evaluated before (for instance in Ólason et al., 2022). We divide the Arctic region into 8 sub-domains. The first 6 are similar to those considered in Ricker et al. (2021) (corresponding to the Barents, Kara, Laptev, East Siberian, Chukchi and Beaufort seas). We also consider 2 additional regions that are the Eastern and Western sectors of the Central Arctic (the sub-domains labelled 7 and 8 in Figure 2, respectively). The Eastern sector is typically covered by first year ice being advected towards Fram Strait following the Transpolar Drift, while the Western sector is mostly covered by multiyear ice, generally thicker (>2m) and less mobile than the ice present in the Eastern sector.

## 3   Observations used for model evaluation

### 3.1   Sea ice concentration, volume and drift

We take sea ice concentration from the climate data record of the EUMETSAT Ocean and Sea Ice Satellite Application Facility (OSI-SAF, Lavergne et al., 2019). To cover the period 2000–2018, we use two different versions of the product: the OSI-450 (1980–2015), and the OSI-430-b (2016-present). Sea ice volume and thickness are evaluated using two independent datasets: outputs from the Pan-Arctic Ice-Ocean Modeling and Assimilation System (PIOMAS, Zhang and Rothrock, 2003) and data produced by combining the observations retrieved from the CryoSAT-2 and SMOS satellites, referred to as CS2SMOS (version 2.2, Ricker et al., 2017). PIOMAS data are the results of coupled ocean–sea-ice model simulations with the daily assimilation of satellite sea ice concentration and sea surface temperature. The main interest of the PIOMAS dataset is that it is available

for the whole simulated period and has been thoroughly evaluated against ice thickness observations (e.g. Schweiger et al., 2011; Laxon et al., 2013; Stroeve et al., 2014), meaning that some of its biases are known. For this reason, it is regularly used as a reference for large-scale sea ice thickness evolution in the Arctic (e.g. Spreen et al., 2020; Davy and Outten, 2020). For evaluating the sea ice drift, we use the low-resolution OSI-SAF sea ice drift product that provides 2-day integrated sea ice displacement (Lavergne et al., 2010). This product includes information about summer ice drift and the uncertainties associated with the drift retrieval from June 2017 onwards.

For each dataset, we process OPA-nex output in order to compare them with observations in a consistent way. First, we integrate in time the 6-hourly OPA-nex output to obtain an output with the same time frequency as the observations (i.e. daily for sea ice concentration, 2-days displacement for the drift, and weekly for the thickness/volume in CS2SMOS). We then interpolate OPA-nex outputs onto the same grid as the observations. There is an additional step for the ice thickness/volume: as PIOMAS is only available monthly, we do a monthly average of OPA-nex outputs and CS2SMOS data, and compare the three datasets with PIOMAS data interpolated on the CS2SMOS 25 km grid.

### 3.2 Sea ice volume and area export through Fram Strait

We also evaluate the ice volume export through Fram Strait, as it is an important contributor to the Arctic sea ice mass balance (Spreen et al., 2009; Ricker et al., 2017; Spreen et al., 2020). We use the time series produced by Spreen et al. (2020), which covers the period 1992–2014, overlapping with most of our simulation. This dataset is based on sea ice thickness measurements derived from 1-4 upward looking sonar measurements installed on moorings across the strait (Vinje et al., 1998) (from which the section of ice thickness is extrapolated), and sea ice drift retrieved from the JPL sea ice motion dataset (Kwok et al., 1998). We also compare OPA-nex results with the time series of Spreen et al. (2009), spanning 2003–2009, that is obtained by combining sea ice thickness from ICESat altimeter observations and sea ice area and drift retrieved from AMSR-E 89 GHz. To compare our export to these datasets in a consistent way, we first estimate the sea ice transport (i.e. motion vectors) at the output frequency of OPA-nex (6h), then interpolate these transports onto the same grid as Spreen et al. (2020). The gate used for the computation is located at $\simeq 79^{o}$N (see Figure 2).

We also compare the simulated sea ice area flux through Fram Strait to the time series estimated by Smedsrud et al. (2017). They combined a blended historical and modern record of sea ice concentration from the National Snow and Ice Data Center (NSIDC, Walsh et al., 2017) with sea level pressure observations across Fram Strait to retrieve the sea ice area flux over the period 1935–2015. Based on the 6-hourly sea ice drift and concentration outputs from OPA-nex, we estimate a time series of the sea ice area flux across the same section at $79^{o}$N (see Figure 2).

### 3.3 Observed contributions to sea ice mass balance

In order to evaluate the ice mass balance and its spatio-temporal variations, we make use of the ice volume change dataset from Ricker et al. (2021). The originality of this dataset is that it separates the dynamic contribution (i.e. the import/export of ice in each region) from the thermodynamic contribution (i.e. the net sea ice growth in winter) to ice volume change in the freezing season (November to March) for 6 Arctic seas (Barents, Kara, Laptev, East Siberian, Chukchi and Beaufort, see Figure 2).

To estimate these contributions, Ricker et al. (2021) combine sea ice volume (Hendricks et al., 2018) and motion information (Girard-Ardhuin and Ezraty, 2012) to retrieve the dynamic volume change, and then subtract it from the total volume change to estimate the net sea ice growth. Ricker et al. (2021) also compare their estimates with outputs from 2 models, PIOMAS and NAOSIM (Kauker et al., 2003). In OPA-nex, we compute the dynamic volume changes in each region from the transports across the gates shown in Figure 2. Thermodynamic volume changes are directly output from the model.

## 4   Model evaluation

As stressed in the introduction, the internal stress is an important term in the momentum equation with the potential to affect the Arctic mass balance, and it is the first time the mass balance of a sea ice model with a brittle rheology is investigated in detail and over a time period longer than a year. This is also the first time such a model is coupled to an ocean. This section therefore focuses on a thorough evaluation of sea ice properties in our simulation to verify that a reasonable Arctic sea ice mass balance is obtained.

### 4.1   Evaluation of simulated sea ice extent, thickness, volume and drift

We first evaluate the large-scale properties of the simulated sea ice. Our computations of bias, RMSE and integrated ice-edge error (IIEE, Goessling et al., 2016) are done in a similar way to Williams et al. (2021) (section 4.1). The evaluation of small-scale dynamics of sea ice in the coupled neXtSIM/OPA setup provided no qualitative differences in sea ice deformations compared to a standalone setup (see Figure A1 and Ólason et al., 2022).

We start our evaluation with the sea ice extent (Figure 3). To quantify the agreement between OPA-nex and the OSI-SAF data over the study domain, we compute the integrated ice-edge error (IIEE), a metric used to evaluate the quality of predicted ice extent that accounts for both errors in the absolute extent and misplacement of ice (Goessling et al., 2016). The IIEE in our study domain is almost zero in winter (December to May), mostly because we limit our analysis to the Arctic Basin, which is fully ice covered in those months. If we extend our analysis to the whole model domain (that also includes most of the North Atlantic, the Hudson Bay and the Baltic Sea, but not the Pacific side), the modelled sea ice extent remains consistent with observations over the winter. The IIEE remains low (0.62M km$^2$ in average in March) and almost constant over the winter, with a small tendency of the model to underestimate the maximum extent. The IIEE increases in summer and peaks in September (1.7M km$^2$ in average for this month), mostly due to misplacement of the summer minimum extent (as the absolute value of the extent is generally well estimated, with the exception of 2016 and 2017). Therefore, OPA-nex successfully captures the seasonal cycle of the ice extent, and its interannual variability, particularly in the summer, with the exception of 2016 and 2017 where the ice loss is overestimated.

We then examine sea ice volume (Figure 4a). Agreement with PIOMAS is generally good, although sea ice volume in OPA-nex is generally lower than in PIOMAS in the early 2000s. After 2008, the agreement becomes very good in both winter and summer, and the two models show a similar interannual variability. A lot of factors could explain the discrepancies between OPA-nex and PIOMAS (differences in atmospheric forcings, in the dynamics and thermodynamics of the models, and the use

of data assimilation in PIOMAS), and it is difficult to attribute these differences to one or the other of these factors. OPA-nex agrees well with CS2SMOS, with an average RMSE of 0.34m and an average bias of 0.03m for the whole period when observations are available (from October to April each year from 2011). This is also true for the ice thickness distribution during the ice growth season (Figure 4b,c,d). Biases in the distribution in OPA-nex compared to CS2SMOS are quite similar to those in PIOMAS and also found in most sea ice models: thick ice is not as thick as the observations in the Central Arctic, and thin ice is often too thick, particularly in the western side of the Arctic Basin (Stroeve et al., 2014; Watts et al., 2021).

The simulated drift generally shows a good agreement with the OSI-SAF data (Figure 5), with a low negative bias (-0.35 km/day on average from 2010 to 2018) and a low RMSE (3.82 km/day) for the freezing season (October to April), when most of the data are available. OPA-nex also captures both seasonal and interannual variability (Figure 5a) and the day-to-day variability (Figure 5b). Uncertainty and drift estimates in the summer only start in June 2017, hence our choice of zooming in on the year 2018 in Figure 5b. From Figure 5b, modelled summer ice drift is overestimated in OPA-nex compared to OSI-SAF, but remains within the larger uncertainties of observations during the melting season. Importantly, the variability remains well-captured all year round.

### 4.1.1 Sea ice export through Fram Strait

Sea ice volume export through Fram Strait is an important term of the Arctic sea ice mass balance. Observations suggest $\simeq$ 13% of the total ice volume in the Arctic Ocean is exported through Fram Strait each year (Spreen et al., 2009; Ricker et al., 2018; Spreen et al., 2020), representing more than 90% of the total sea ice export out of the Arctic (Haine et al., 2015). Figure 6a shows that OPA-nex captures the observed export well as it remains within the standard deviation based on daily transport values estimated by Spreen et al. (2020) over the studied period. However, the model tends to underestimate the magnitude of the export, particularly before 2008. The variability of the export is captured generally well, with a detrended determination coefficient of $R^2 = 0.60$. Again the model seems to perform better after 2008 ($R^2 = 0.70$). The underestimation of the sea ice export before 2008 has an important consequence when we examine the sea ice export trend: while Spreen et al. (2020) find a decreasing trend in the export, we find no significant trend in OPA-nex. It is therefore interesting to investigate the reason behind this discrepancy.

Sea ice export depends on 3 quantities: thickness, velocity, and concentration across the section. We first examine these two latter quantities by comparing the sea ice area flux in OPA-nex to the time series from Smedsrud et al. (2017) over the period 2000–2015 (Figure 6b). OPA-nex captures this area flux very well (RMSE= $20.28 \times 10^3 \text{km}^2/\text{month}$, $R^2 = 0.81$), which suggests that the model successfully reproduces both the extent of ice in Fram Strait and the ice drift over this period. The discrepancy between our export and the one from Spreen et al. (2020) therefore likely comes from a difference in sea ice thickness across the section, hinting that OPA-nex does not get thick enough ice in the Fram Strait prior to 2008, which is a typical bias in sea ice models (Watts et al., 2021). We note that OPA-nex shows a better agreement with the observations of Spreen et al. (2009), which highlights the uncertainties associated with methods used to retrieve the ice thickness along the section. Sea ice export in 2005–2006, however, remains underestimated in OPA-nex. This, and the fact that the ice volume in OPA-nex for this period is in general lower compared to the PIOMAS model (Figure 4a) suggests OPA-nex sometimes

underestimates sea ice thickness over the period 2002–2008. The period 2007–2008 corresponds to a large loss of old ice in the Arctic (Kwok, 2018), which suggests that this underestimate could be due to a negative bias in the thickness of the older ice prior to 2008 in the model.

### 4.1.2 Regional winter ice mass balance

We now investigate the dynamic component of the ice mass balance (the net balance between import and export of sea ice) within the Arctic Basin, in a similar way to Figure 3a of Ricker et al. (2021). The ice transport contribution to the mass balance for the regions where data is available is very well estimated in OPA-nex (Figure 7). The variability is well captured, with determination coefficients generally exceeding 0.50 between the detrended OPA-nex results and estimates from observations. Similarly to Ricker et al. (2021), we do not find any significant trend over the period 2002–2018 (note that they also include 2019 in their study) for any of the regions analysed here. This is also true for the Central Arctic regions (West and East) that are not included in Ricker et al. (2021).

Figure 8 shows the same analysis but for thermodynamic processes, comparable to Figure 3b of Ricker et al. (2021). The thermodynamic processes included in OPA-nex are the production and growth of young ice, the basal growth of (thicker) ice, and ice formed due to the flooding of snow, as well as basal and surface melt. The magnitude of the net winter growth is estimated well in general. The main discrepancies between OPA-nex and Ricker et al. (2021) are found in the Kara and East Siberian seas, where OPA-nex overestimates the amount of ice formed every winter. This overestimation of ice growth in these seas is also visible in the data from PIOMAS, shown as a reference in Ricker et al. (2021). As in Ricker et al. (2021), we find a small but statistically significant (i.e with a $p$-test result lower than 0.05) decreasing trend in ice production in the Kara Sea. Yet, in contrast to their study, we do not find any significant increase in the Chukchi Sea. All other regions are found to have insignificant trends in both OPA-nex and estimates by Ricker et al. (2021). Interannual variability of the net ice growth in each region is generally significantly smaller than the one of the net transport (by a factor $\simeq 2$ for regions 5 to 8), and is not well captured by OPA-nex.

The comparison with Ricker et al. (2021) suggests that the ice mass balance in OPA-nex is captured well in winter. In the next section, we analyse the Arctic sea ice mass balance but this time for the whole Arctic Basin, without limiting ourselves to periods covered by observations.

## 5 Arctic sea ice mass balance

### 5.1 Contributions of thermodynamic and dynamic processes

We quantify the contribution of each source and sink of sea ice over time in the domain of interest (Figure 2) for the whole study period (Figure 9a,b). As in section 4.1.2, we partition the sources and sinks of sea ice into dynamic and thermodynamic processes. The dynamic processes are sea ice transport through Fram Strait (in green) and the sum of ice transport through all the other external boundaries of the domain (Figure 2). The thermodynamic processes are the same as in Figure 8. This way of

presenting the mass balance is similar to what Keen et al. (2021) have done for sea ice components of climate models that are part of the latest Coupled Model Intercomparison Project (CMIP6). The only difference is that the process we call "growth of young ice" includes both the ice volume of frazil ice production and the ice volume corresponding to the growth of this frazil ice until it enters the consolidated thicker ice category, which occurs when the thickness of the young ice exceeds $h_{max}$ (see section 2.1 for details). This definition of young ice is broadly similar to the one of the World Meteorological Organization (forming ice thinner than 30 cm, WMO, 2014).

We find that the interannual variability of the net mass balance in our domain is controlled by both thermodynamics and the export through Fram Strait, these two terms being of similar amplitude (Figure 9a and Figure 10). Sea ice transport through other gates is negligible compared to the export through Fram Strait (hence almost not visible in Figure 9a,b). Previous reports suggest that Fram Strait represents $\simeq$90% of the net sea ice export of the Arctic, the second main source of export being through Davis Strait, south of our domain (Carmack et al., 2016). In our case, the contributions from all gates other than Fram Strait almost cancel out, being slightly positive (ca. $+1 \times 10^3 \text{km}^3$ over 2000–2018, against around $-30 \times 10^3 \text{km}^3$ through Fram Strait). This is likely because i) the Canadian Arctic Archipelago is not included in our analysis and we therefore miss all the ice that forms there and is then exported through Davis Strait, and ii) 12km is too coarse to resolve the outflow through Nares Strait, leading to an underestimated export through this narrow gate (only $\simeq$1km$^3$/year in the model, while observations suggest an average up to $\simeq$190km$^3$/year over 2017–2019, Moore et al., 2021). If we consider the seasonal cycle of Arctic sea ice volume, sea ice export only plays a minor role in the variations of the ice volume, which are mostly driven by sea ice thermodynamics (Figure 9b). We note that this seasonal cycle is very similar to the multi-model mean seasonal cycle presented in Keen et al. (2021) (see their Figure 4a) even though the framework (fully-coupled climate models, 1960–1989 climatology) is different.

Sea ice production slightly exceeds melt in the domain (with the exception of 2016, when the net ice production becomes negative; Figure 9a and Figure 10). This is because we have excluded the domain south of Fram Strait, where a large part of the melting occurs (Figure 9d). The yearly amount of ice growth in the domain is closely linked to the amount of melt (Figure 9a). This is most likely because strong melt events lead to large areas of open water and thinner ice at the end of the summer, enhancing the refreezing in the next autumn and winter (Petty et al., 2018). We do not find any trend in sea ice growth nor melt using this domain, and large changes in the total ice volume (as in 2002, 2012, 2014 or 2016) are mostly associated with the interannual variability of the balance between melt and growth (Figure 10).

Overall, interannual variations of the net volume change associated with thermodynamic processes are mostly due to variations in the basal growth of thick ice and basal and surface melt (Figure 9a). Basal and surface melt contribute about equally to the yearly ice melt in the domain, while the basal melt dominates south of Fram Strait (outside the study domain), likely because sea ice encounters warmer surface waters in the Greenland Sea (Bitz et al., 2005; Lei et al., 2018). Young ice growth accounts for about half of the yearly ice production. This proportion is sensitive to the choice of minimum and maximum thickness for the young ice in our 3-category thermodynamics scheme in OPA-nex (2.1), as is the case in most models (Keen et al., 2021). Young ice growth variability is weaker than that of basal growth, and is mostly controlled by two drivers: the ice extent at the end of the summer (positive anomalies are found in e.g. 2008, 2013 or 2017, which are years following low-extent

anomalies), and the amount of openings (leads or polynyas) that are present in pack ice. Ice production due to the flooding of snow is negligible for the area and time period discussed in this analysis. This contribution may, however, be underestimated by the mass-conserving snow-ice formation scheme used (Turner et al., 2013). The ice volume loss over time in the domain is clearly visible in Figure 10. This loss is qualitatively similar to the winter volume evolution reported by Liu et al. (2020) using ice age to estimate the ice volume. We find a statistically significant ($p \simeq 0.01$) trend of -280 km$^3$ per year over 2000–2008, which is within the range of sea ice volume trends (from both models and observations) discussed in Liu et al. (2020) (between $\simeq -200$ km$^3$ and $-400$ km$^3$), but no significant trend for the period 2009–2018 (also as reported in Liu et al., 2020). The comparable orders of magnitude of the ice exported and the net change in ice volume due to thermodynamics reinforce the importance of the dynamic contribution to the mass balance that was suggested in Figure 9. Net ice production peaks in 2013–14, when it dominates the net export by a factor of $\simeq 2$, resulting in an increase of the ice volume in the domain. Loss of sea ice volume in the domain mostly occurs in years of low (or negative) net ice production (such as 2002, 2012 and 2016). The yearly net sea ice export varies very little in comparison to the net sea ice production.

## 5.2 Contributions of leads and coastal polynyas to winter ice production

We now estimate the contribution of leads and polynyas to the winter ice mass balance. This estimate is based on the simulated ice formation in open water and ice growth in the young-ice category (see section 2.1). In winter and in pack ice, such ice growth will only take place where the ice has been recently diverging, because young ice quickly grows thick enough to be transferred to the "old ice" category (a few days at most). In the absence of divergence, the domain would be fully covered by old ice. The following analysis could be carried out with any sea-ice model with multiple ice thickness categories. However, the amount of ice produced in openings (i.e. leads and polynyas) in pack ice and its localisation are very likely to be strongly impacted by the ability of the model to reproduce the small-scale sea-ice dynamics. This is because the highest values of divergence rates (and deformation rates in general) in Arctic pack ice are very localised (Figure A1a,b), which would not be the case if the ice cover was homogeneous (e.g., Stern and Lindsay, 2009). For instance, Bouillon and Rampal (2015) found that in neXtSIM at 10 km resolution, 50% of the divergence in the Central Arctic was associated with only 5-10% of the surface area in the domain used for the analysis (this surface ratio would be 50% in the case of a homogeneous ice cover). Divergent ice motion, therefore, results primarily in the formation of localised leads in the central pack or of polynyas near the coast. An underestimate of divergence rates, which "standard" sea ice models run at resolutions coarser than 5km tend to do (Hutter et al., 2022), would therefore imply a subsequent underestimation of ice production in winter if there is not a sufficient parameterization to represent the effect of leads. This parameterization can be done using, for instance, a minimum value for the lead fraction in each grid cell, resulting in a more uniform distribution of lead growth over the domain (as this can be done in the LIM3 model, Rousset et al., 2015). The importance of resolving leads versus using parameterizations to represent the ice growth in leads in numerical models has not been assessed to our knowledge. This would likely require a model comparison between a model which captures divergence rates well and another one using a parameterization for leads, which is out of the scope of this study. Instead, we focus on estimating the importance of ice production in leads in our simulation, as this has not been estimated at a Pan-Arctic scale before. The advantage of using neXtSIM in our analysis is that its ability to reproduce

small-scale sea ice dynamics has been thoroughly evaluated before (see Ólason et al., 2022, and appendix A). In addition, it has been shown that the model is able to capture rates of divergence consistent with observations and relevant statistics of the observed lead fraction in the Central Arctic at spatial resolutions similar to the one used here (Ólason et al., 2021, 2022, and Figure A1).

The impact of leads and polynyas on winter ice production is visible in Figure 11a, and is clearly linked to the growth of young ice (Figure 11b). We note that the spatial patterns in Figure 11a,b look similar to maps of observed ice divergence or lead fraction distribution described in previous studies (e.g. Kwok, 2006; Willmes and Heinemann, 2016; Wang et al., 2016; Zhang et al., 2018). The imprint of leads on ice production is particularly visible in the Beaufort Sea, with long linear features orthogonal to the coast. Their presence in a 18-year long climatology demonstrates their strong impact on sea ice production and the likely recurrence of these features year after year. Lead-type features are also visible in the Central Arctic when looking at the contribution of openings to the total growth (Figure 11b). This is likely because the thick ice covering the Central Arctic limits the amount of ocean heat loss, hence basal growth, that can occur, meaning that local openings in this thick ice cover significantly contribute to the total ice growth there. We also note that coastal areas are places of intense production of newly formed ice in winter, likely due to the recurrent opening of coastal polynyas. Before quantifying the impact of leads and polynyas to winter ice production, we assess the limitations of associating the growth of young ice with these types of features. From Figure 11c, we can differentiate 2 phases in ice growth within the freezing season. The first period (October to December) is when ice production is dominated by the growth of young ice from open water, as the open water refreezes. Young ice growth occurs in open water areas in a homogeneous way until the whole Arctic Basin is ice covered. The second phase is from January to March, when the contributions of basal growth and young ice growth reach an equilibrium. This corresponds to a plateau in the contribution of the growth of young ice to total growth, visible every year in Figure 11c. Sea ice concentrations in the domain are then very close to 100% everywhere, which means that the young ice is mostly (if not totally) produced from openings due to divergence in pack ice (Strong and Rigor, 2013). There are, however, regions in our domain that remain covered with thin ice in winter, or even include open water areas (e.g the Barents Sea). To avoid the inclusion of young ice production in MIZs in our analysis, we remove the southern part of the Chukchi Sea, as well as the Barents and Kara seas from the domain we consider (black contour, Figure 11a,b).

We now quantify the contribution of ice growth due to openings within pack ice over the total ice growth. This contribution adds up to $\simeq$25% to 35% of the ice growth in the domain (black line in Figure 11e). This corresponds to an annual winter ice volume production of $\simeq$270 to 380 km$^3$ (black line in Figure 11d). We find that young ice growth is increasing over the studied period with a significant (i.e with a $p$-test result lower than 0.05) positive trend of +74.7 km$^3$ per decade. This is also true for its contribution to total growth, with a positive trend of + 4.3% per decade. Over the same period, the basal and total growth show no significant trends (not shown).

As mentioned before, ice growth from leads and polynyas is particularly substantial in coastal areas. It is therefore interesting to distinguish between coastal areas, where both leads and polynyas can occur, from the interior Arctic Basin, where most openings correspond to leads. We reproduce our analysis, excluding this time the regions shallower than 300m (cyan dashed contour in Figure 11,a,b). We find that the interior Arctic basin accounts for about half of the young ice growth from openings

in pack ice in winter (grey lines in Figure 11d,e). We still find significant positive trends in young ice growth (+29.4 km$^3$ per decade) and its contribution to total ice growth. This means that while ice production in leads in the interior Arctic is a significant contributor in the study domain, it is the coastal areas that are most important to the modelled increase in ice growth in leads and polynyas over 2000–2018 (Figure 11d).

We note that most of the increase in the contribution of ice growth in leads and polynyas to total ice growth takes place from 2000 to 2010, with a significant positive trend of about +7% over this decade (whether the shelves are excluded or not). From 2008 to 2018, we find no significant trend for ice growth in leads and polynyas. Using 2018 as the last year of the analysis, the positive trend in the contribution of ice growth in leads and polynyas to total ice growth is only significant when starting prior to 2005 (included), and prior to 2002 if we exclude the shelves. This highlights the important contribution of the earlier years of the analysis (the period from 2000 to 2008 in particular) to this increasing trend in ice production.

## 5.3 Regional variability and trends in winter ice production in leads/polynyas

To better understand this evolution of ice growth in leads and polynyas, we now examine each region included in our analysis (Figure 12a-f). Although ice growth in leads and polynyas strongly varies from one region to another (Figure 12a), its contribution to total growth is similar across the subdomains, around 30% if we account for coastal areas, and about 25% when we only consider the interior of the basin (Figure 12e). However, there are two regions where the behaviour differs from the other regions. The contribution is lower in the Western Central Arctic ($\simeq$24% including coastal areas), likely due to thick sea ice, which is not very mobile (Figure 12g,i). Oppositely, the Eastern Central Arctic is characterised by a large contribution of leads and polynyas to sea ice production ($\simeq$44%; $\simeq$39% when excluding the shelves), likely due to the large cracks that regularly form as the ice undergoes high internal stresses while exiting through the narrow Fram Strait (as visible in Figure 11a, and in lead frequency maps in Willmes and Heinemann, 2016). Increases in winter ice volume growth due to leads and polynyas (and their contribution to total growth) occur in almost every region over 2000–2018 (Figure 12b,f), but are only significant in the regions along the Eurasian Coast (Laptev, East Siberian and Chukchi seas), and in the interior of the Chukchi Sea. The largest trends are found in the Laptev and East Siberian seas, as well as in the interior of the Chukchi Sea. Total ice volume growth shows no significant trend anywhere (Figure 12d).

We now try to relate this increase in the ice growth in leads and polynyas to two other sea ice quantities that are related to sea ice deformation: sea ice drift and thickness (Figure 12g-j). Sea ice drift speed is found to increase over time, with positive significant trends varying between +10% to +20% per decade depending on the region (Figure 12h). Spatial variability of sea ice drift trend magnitudes follows that of ice growth in leads and polynyas, which suggests a close relationship between the two. As for the ice growth in leads and polynyas, most of this increase in ice drift speed occurs over the period 2000–2008, there is no significant trend for any region for the period 2008–2018 (not shown). In the case of sea ice thickness, the relationship with ice growth in leads and polynyas is less clear. The thinning trend is significant almost everywhere (Figure 12j), but the spatial distribution of the trend magnitude does not reflect the one of ice growth in leads and polynyas. We note that sea ice drift and thickness are not independent, and the increase in ice drift speed is most likely driven by the thinning of the ice and the associated reduction of the ice strength (Rampal et al., 2009). Similarly to what was found in previous studies (Rampal

et al., 2009; Kwok et al., 2013), we find no or very little trend in the wind speed, with magnitudes that are too low to explain the increase in ice drift (not shown).

Our interpretation of the results is that the increasing trend in the ice drift velocity is associated with higher divergence rates, enhancing winter ice production in openings in the ice pack. This feedback has been suggested before. Kwok (2006), for instance, has hypothesised that it could contribute to the resilience of sea ice in the Arctic. This enhanced winter ice production is particularly intense close to the coast and in regions with thin ice, such as the Laptev and Siberian seas (Figure 12b,f). These regions are also associated with rather low average ice drift speed in winter, most likely due to the presence of landfast ice. Therefore, landfast ice, and the leads and polynyas that form along it, likely play an important role in the production of ice in winter along the Eurasian coast.

## 6  Discussion

In our analysis, we have highlighted the importance of the contribution of sea ice dynamics to the sea ice mass balance. One interesting result is our estimation of the winter ice growth that is associated with leads and polynyas. This quantity has, until now, not been estimated at the pan-Arctic scale due to difficulties in estimating it from observations, particularly on large scales, and the under-representation of LKFs by most models for spatial resolutions larger than $\simeq 5$ km (Hutter et al., 2022; Bouchat et al., 2022). At first glance, our estimate that sea ice production in leads contributes to between 25 and 35% of the winter ice production in the Arctic agrees well with previous estimates from Kwok (2006, $\simeq 25$ to 40%) and von Albedyll et al. (2022, $\simeq 30$%). However, we acknowledge that the methods, time periods, and the spatial and temporal scales we use are different to the one used in these two studies. In the following paragraphs we briefly discuss these values and their context, although a fully consistent comparison with each of these estimates remains outside the scope of this study.

In their study, von Albedyll et al. (2022) estimate that ice production in leads represents about 30% of ice production in the 2019–2020 freezing season, but they distinguish two periods: from October 2019 to early April 2020, when ice formation in leads contributes to around 10% of sea ice production, and from April 2020 to June 2020, when most of the ice formation takes place in leads and the net contribution of basal growth is almost zero. Their estimation gives a Lagrangian view of ice production in pack ice, following the drift of the MOSAIC expedition between the north of the East Siberian Sea to the north of Svalbard. Their estimate of 10% for the period from October to early April is not very different from our estimate for the East Siberian Sea when excluding the shelf coast ($\simeq 18$%, Figure 12e). This is consistent with the fact that a large part of the vessel drift took place north of $85^{o}$N, where the contribution of leads to winter ice production is rather low (generally $\leq 20$%, Figure 11b). It is also likely that, in autumn, basal growth contribution to ice production in pack ice is larger than from January to March, as level ice and its snow layer are generally thinner, hence allowing for more heat loss from the ocean in ice covered areas. von Albedyll et al. (2022) suggest that the higher contribution of leads to ice production in spring than in winter could be partly resulting from regional differences, as the vessel drifted towards regions with a higher contribution of leads to the ice production. Our model suggests this is likely the case (Figure 11b).

Kwok (2006) investigates the western part of the Arctic Basin over the period 1997–2000, when most of the ice cover consisted of multiyear ice. To estimate the quantity of ice that is produced in leads, they combined a thermodynamic model with sea ice deformations retrieved from the Radarsat Geophysical Processing System (RGPS, Kwok et al., 1998). They used a rough estimate of the amount of basal and total growth to provide an order of magnitude of the contribution of leads to the total ice production from November to April. In our analysis, we excluded the late autumn (November-December) to avoid including frazil production in MIZs, and focus on young ice growth in pack ice, associated with leads and polynyas. However, in the late 1990s sea ice was already compact in the western part of the Arctic Basin from November onwards. Therefore we can estimate the contribution of leads and polynyas to the ice growth from November 1999 to March 2000 in OPA-nex using the same method as in section 5.3. We find values of 30% for the Western Central Arctic and 35% for the Beaufort Sea (22% and 25% respectively when excluding the shelf area), which is within the range of values estimated in Kwok (2006).

Our estimates of ice growth in leads are mostly sensitive to i) rheological parameters affecting sea ice divergence and ii) the maximum thickness of young ice ($h_{\mathrm{max}}$). In case i), we ensured that our results were consistent with the stand-alone version of neXtSIM presented and evaluated in Ólason et al. (2022), which was found to produce realistic divergence rates. In case ii), we tested $h_{\mathrm{max}}$ values in the range [12.5cm, 22.5cm], and found that larger values resulted in overestimated ice thickness, particularly in the Kara and Barents seas. This range of values seems reasonable to represent the transition between forming ice (frazil, pancake, nylas) and consolidated first-year ice.

To our knowledge, the increase in both the amount of ice produced in leads and its relative contribution to the winter ice production has not been reported before. Studies focusing on leads in the Arctic often investigate the evolution of the observed lead frequency (Lewis and Hutchings, 2019; Willmes and Heinemann, 2016) or the modelled lead area fraction (Wang et al., 2016; Ólason et al., 2021). A consistent comparison of the lead frequency between OPA-nex and observations is not straightforward, but it is likely that lead frequency is closely linked to the amount of ice production in leads. Willmes and Heinemann (2016) investigate the evolution of lead frequency in the Arctic over 2003–2015 using thermal infrared imagery. They find that interannual variability is large and there is no significant trend. If we limit our analysis to 2003–2015, we also find no significant trend in the evolution of ice growth in leads and polynyas (Figure 11d). Further analysis of observations and model results is therefore required to further investigate these findings.

## 7 Conclusions

In this study, we have presented a new ocean–sea-ice coupled model and evaluated its representation of the sea ice mass balance over the Arctic region. For the first time in this type of study, the sea ice model uses a brittle rheology to represent the sea ice mechanics. The simulation captures very well the standard sea ice evaluation metrics of sea ice extent and volume, as well as large-scale drift and export through Fram Strait. The winter mass balance is consistent with observations for the period 2003–2018. We estimate the contribution of leads and polynyas to the winter mass balance. This contribution adds up to 25% to 35% of winter ice volume growth, in line with previous estimates from Kwok (2006) based on satellite observations. We also find that this contribution has increased over 2000–2018, mostly due to an increase of openings in coastal areas associated

with an increase of ice drift velocity. Future studies will focus more precisely on the representation of leads in the model, and compare them to available observations (e.g. Willmes and Heinemann, 2016; Reiser et al., 2020) to assess the nature of this increase. Extending the analysis based on the distinction between leads and pack ice over all seasons could also provide new insights into the importance of small-scale dynamics to the ice mass balance, especially as von Albedyll et al. (2022) noted the importance of ice formation in leads from April to June.

*Data availability.* The OSI-450 sea ice concentration product is available at ftp://OSI-SAF.met.no/reprocessed/ice/conc/v2p0 (last visited September 2021). The OSI-430-b sea ice concentration product is available at ftp://OSI-SAF.met.no/reprocessed/ice/conc-cont-reproc/v2p0 (last visited September 2021). PIOMAS outputs are available at http://psc.apl.uw.edu/research/projects/arctic-sea-ice-volume-anomaly/data/model_grid (last visited August 2021). CS2SMOS sea ice thickness product is available at ftp://ftp.awi.de/sea_ice/product/ (last visited September 2021). Low-resolution daily sea ice drift product from OSI-SAF can found atftp://osisaf.met.no/archive/ice/drift_lr/merged (last visited September 2021). Monthly outputs of all quantities discussed in the manuscript are available on zenodo as netcdf files. We also share the data used for each figure, also as netcdf (https://doi.org/10.5281/zenodo.7277523). The neXtSIM code is still in development and will be made open source in the coming months (in a dedicated publication).

## Appendix A: Tuning the coupled model

Coupling neXtSIM to OPA has required the modifications of some of the sea ice model parameters in order to obtain sea ice extent, drift, thickness and deformations that compare reasonably well against observations. In this section, we briefly describe the methodology we followed.

One of the main difference between the stand-alone setup of neXtSIM (used for instance in Ólason et al., 2022), is that in stand-alone, the model is forced using geostrophic currents (in practice, ocean currents at 30 m depth from a reanalysis), whereas in a coupled mode, neXtSIM receives direct information from the surface currents in OPA. This change likely affects the energy and momentum balance at the ice-ocean interface, potentially requiring changes to the values of sea ice parameters affecting the ice drift and deformations. Therefore, we started our tuning with the ice drift, using the years where observations are available in the summer to estimate a range of ice-ocean and ice-atmosphere drag coefficients for which the model ice drift remains within the uncertainties of observed values, assuming that the ice was in free drift (as internal stress is almost negligible in the summer). Choosing one pair of value for these coefficients, we then investigated the rheological parameters $P$ (the scaling parameter for the ridging threshold) and the cohesion of sea ice at the lab scale ($c_{\text{lab}}$). The effects of changing these parameters are described in detail in Ólason et al. (2022). In short they can affect the ice drift (in winter), the spatial distribution of sea ice thickness and the small-scale sea ice deformations. For instance, decreasing $P$ tends to increase the ice drift speed and the gradient of sea ice thickness from the coast of Greenland (where most of the thick multiyear ice is found) to the Eurasian coast (where sea ice is mostly thin first-year ice). Ólason et al. (2022) also show the qualitative evolution of deformations patterns in the Arctic Basin, finding that these patterns look similar to observations by the Radarsat Geophysical Processing System (RGPS, Kwok et al., 1998) in the range [6-14] kPa. We found that this range had been shifted down in the

coupled ice-ocean system compared to the stand-alone neXtSIM setup used in Ólason et al. (2022), with values down to 3 kPa giving probability density functions that match the ones obtained with RGPS and spatial distribution of deformations that look qualitatively similar to RGPS observations (see for instance the case of divergence in Figure A1). We found that P=3 kPa and $c_{\mathrm{lab}}$=2 MPa was a good compromise between thickness distribution (Figure 4), sea ice drift (Figure 5) and deformation patterns (Figure A1).

We also modified some values associated with the thermodynamics. The value maximum thickness of the young ice category has a strong effect on the slope of the sea ice volume evolution during the autumn growth. We chose a value that was giving slope similar to ice volume growth as estimated in CS2MOS. The values for the snow and sea ice albedos were chosen, in their physical range, to give a reasonable match between modelled and observed sea ice extent and thickness.

*Author contributions.* PR, CL and EO obtained the funding. PR, EO and GB formulated the study. LB, CT and EO developed the coupling framework. GB produced the simulation and carried out the analysis. HR helped with analysis. RR provided data for model evaluation. GB wrote the manuscript with input from all authors.

*Acknowledgements.* This research has been funded by the Norwegian Research Council (Nansen Legacy: grant no. 27673, FRASIL: grant no. 263044, and ARIA: grant no. 302934), JPI Climate and JPI Oceans (MEDLEY project, under agreement with the Norwegian Research Council, grant no 316730), and by Copernicus Marine Environment Monitoring Service (CMEMS) WIzARd project. CMEMS is implemented by Mercator Ocean in the framework of a delegation agreement with the European Union Copernicus Marine Environment Monitoring Services (contract no. 69), and the European Space Agency through the Cryosphere Virtual Laboratory (CVL, grant no. 4000128808/19/I-NS). We thank Mathieu Plante and the anonymous referee for their constructive comments and their suggestions which helped us improve the manuscript.

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

**Table 1.** Main parameters used for the sea ice model in this study. All other parameters can be found in Ólason et al. (2022) (for the dynamics) and Rampal et al. (2019) (for the thermodynamics).

| Parameter | symbol | former value (stand-alone) | new value (coupled) |
|---|---|---|---|
| Ice–atmosphere drag coefficient | $C_a$ | $2.0 \times 10^{-3}$ | $1.6 \times 10^{-3}$ |
| Ice–ocean drag coefficient | $C_w$ | $5.5 \times 10^{-3}$ | $6.7 \times 10^{-3}$ |
| Scaling parameter for the ridging threshold | $P$ | 10 kPa/m$^{3/2}$ | 3 kPa m$^{-3/2}$ |
| Main model time step | $\Delta t_m$ | 900 s | 450 s |
| Time step for sea ice dynamics solver | $\Delta t$ | 7.5 s | 6 s |
| Maximum thickness of newly formed ice | $h_{\max}$ | 27.5 cm | 18 cm |
| Sea ice albedo | $a_{\mathrm{ice}}$ | 0.63 | 0.57 |
| Snow albedo | $a_{\mathrm{snow}}$ | 0.88 | 0.8 |
| Critical thickness parameter for ice grounding | $k_1$ | 10 | 5 |

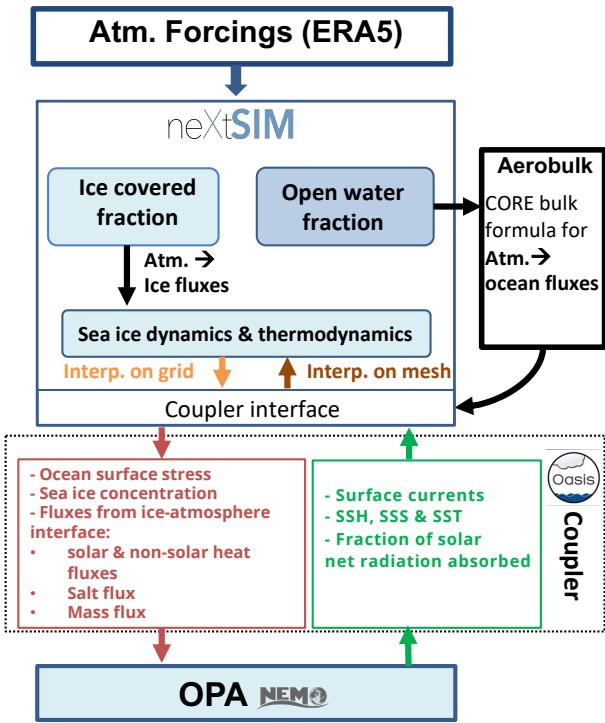

**Figure 1.** Summary of the coupling between neXtSIM and OPA, the ocean component of the NEMO modelling framework.

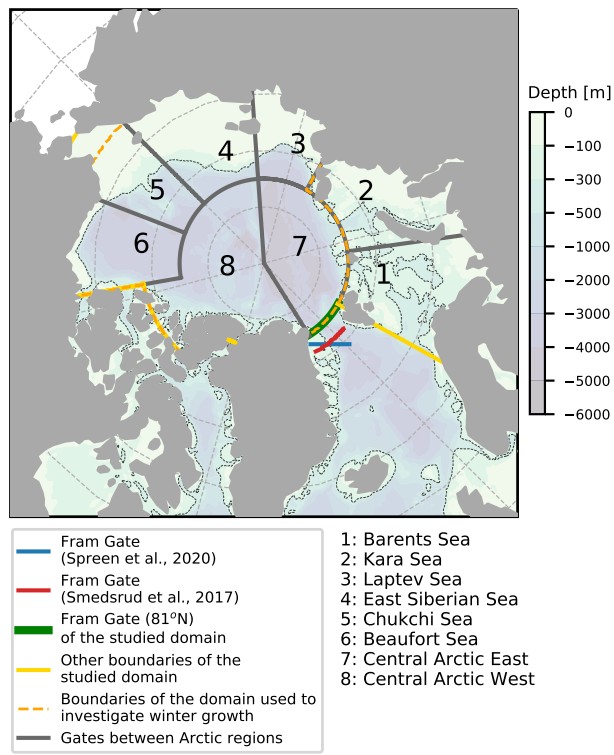

**Figure 2.** Domain, sub-domains and gates used for the analyses presented in this study.

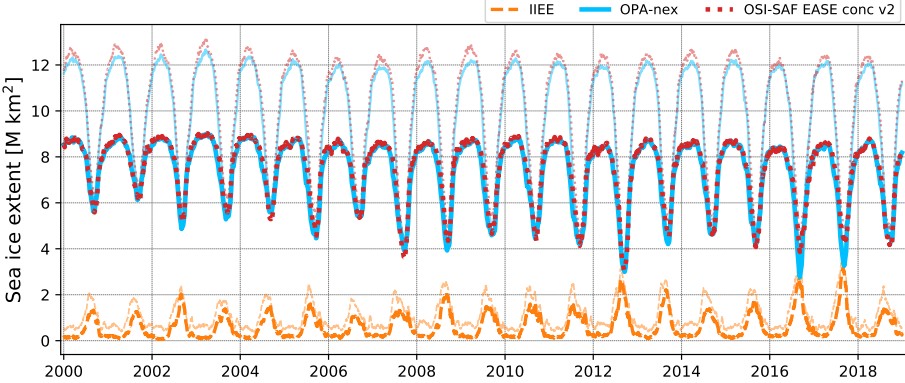

**Figure 3.** Time series of the sea ice extent for OPA-nex (in blue) compared to observations from the OSI-SAF EASE dataset (in red) for the study domain (bright lines) and for the whole domain model (faded lines). The orange dashed line represents the integrated ice-edge error (IIEE, Goessling et al., 2016) for the study domain (bright line) and for the whole domain (faded line).

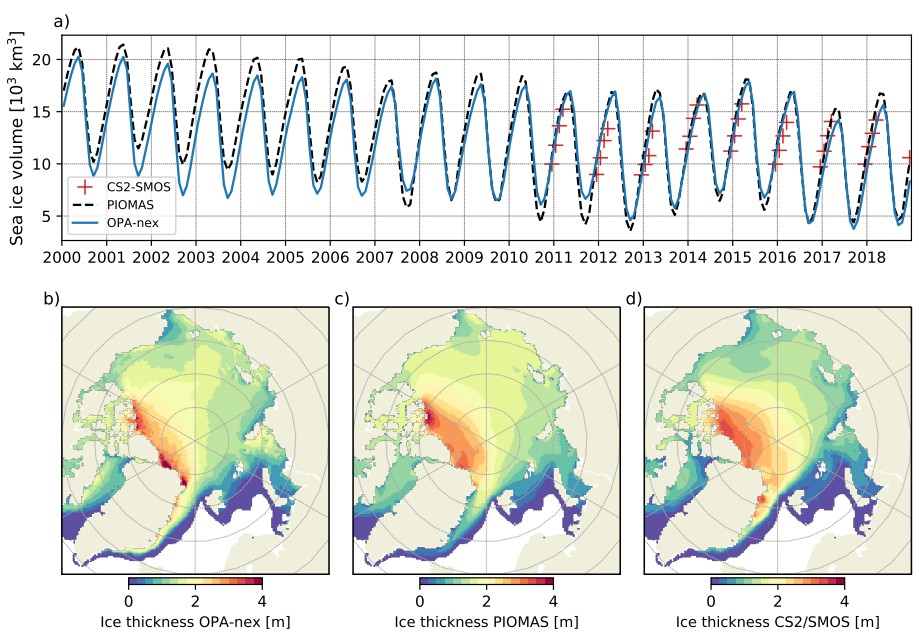

**Figure 4.** Time series of the monthly averaged sea ice volume (a) for OPA-nex compared to other reference datasets (PIOMAS and CS2SMOS). Panels (b, c, d) show a climatology of the sea ice thickness distribution in OPA-nex (b), the PIOMAS model (c) and the CS2SMOS dataset (d). This climatology is computed for the period from December to March over the years 2011–2018 (period of availability of the CS2SMOS product).

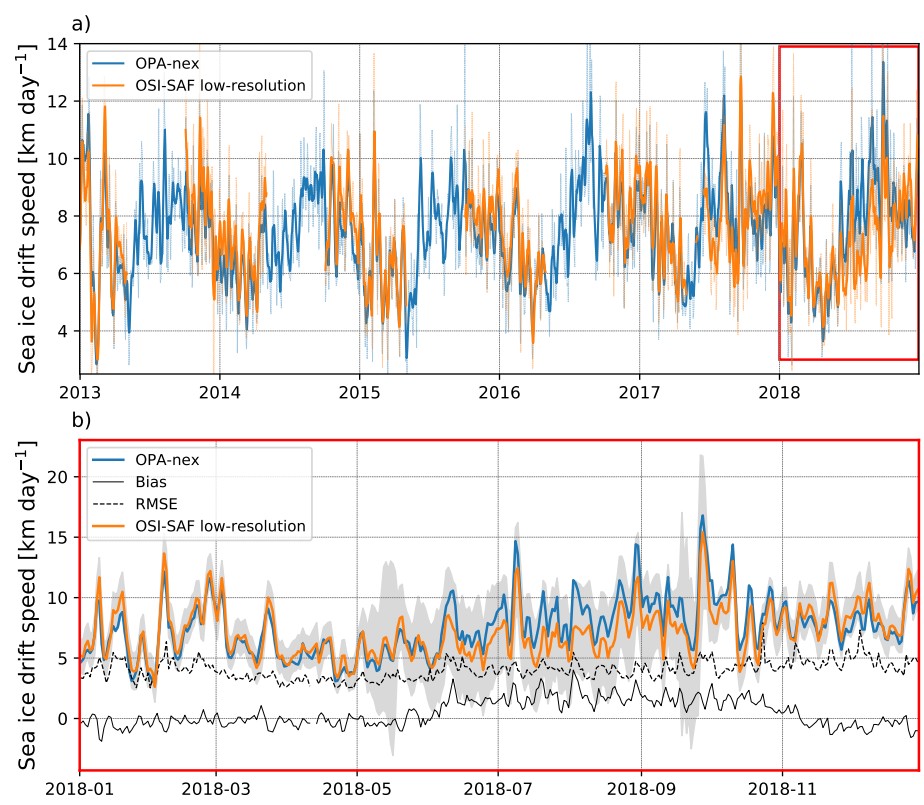

**Figure 5.** (a) Evolution of the spatially-averaged sea ice drift speed for OPA-nex (in blue) and the OSI-SAF low-resolution product (in orange). The OSI-SAF low-resolution product is available from 2010 onwards, but for the sake of readability we only show the period 2013–2018 . The thin lines correspond the daily values and the thick lines correspond to their associated 7-day running average. The red box in (a) delimits the time period over which we show a zoom on the daily values in (b). The solid and dashed black lines in (b) represent respectively the bias and the RMSE between OPA-nex and observations. The shaded area corresponds to the uncertainty of the observations (provided by OSI-SAF).

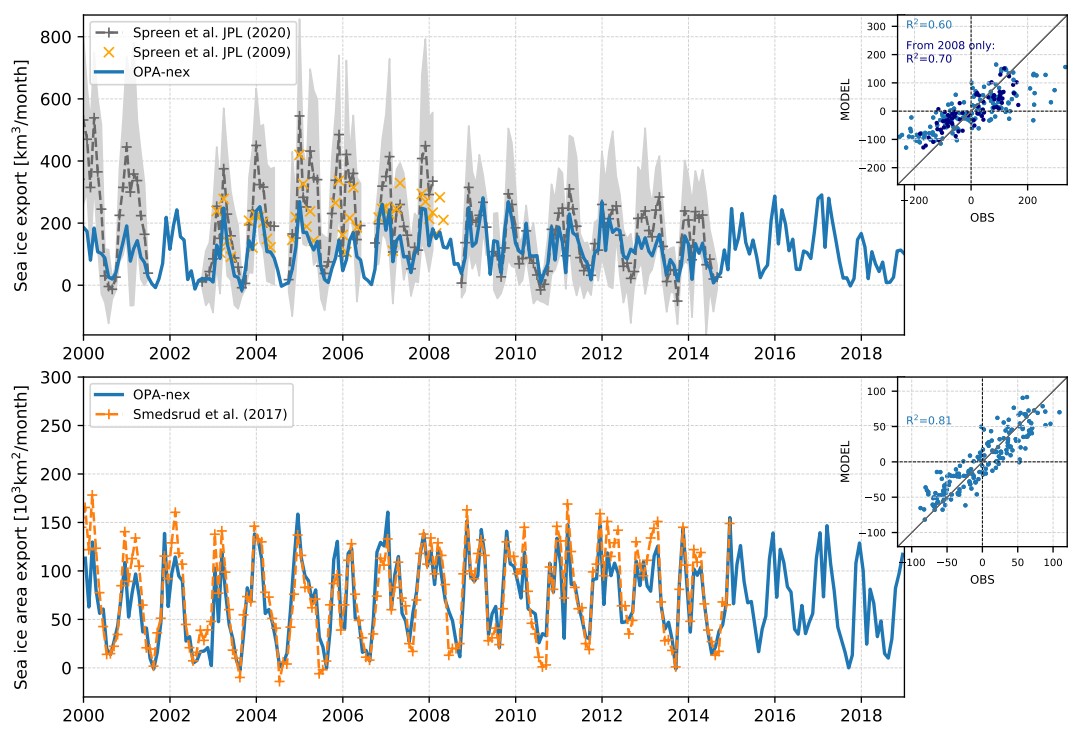

**Figure 6.** Top: Monthly sea ice volume export through Fram Strait ( $79^{o}$N) for the simulation (blue solid line) and as estimated by Spreen et al. (2020, grey dashed line) . The shaded area corresponds to the standard deviation based on daily transport values.

Bottom: Monthly sea ice area export through Fram Strait ( $79^{o}$N) for the simulation (blue solid line) and as estimated by Smedsrud et al. (2017) (orange dashed line). In the top right corner of each graph, the scatter plot shows the correlation between OPA-nex and the associated reference dataset after detrending both time series. $R^2$ correspond to the determination coefficient after 2008 only.

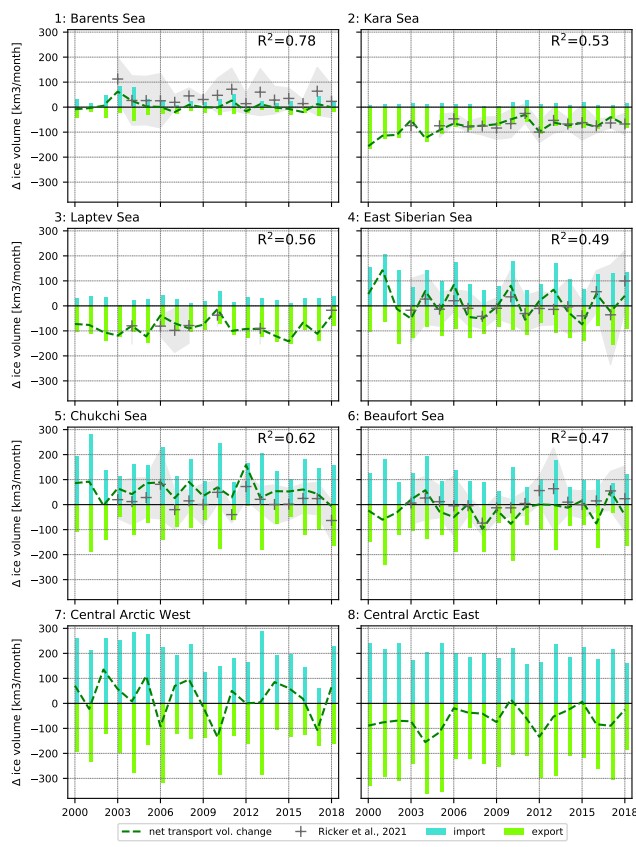

**Figure 7.** Temporal evolution of the sea ice mass balance due to dynamic processes. Results are presented region per region for the winter months (November to March) in OPA-nex (black dashed line) and compared to Ricker et al. (2021) estimates (grey plus signs). The contributions of sea ice export (green) versus import (turquoise) for each domain are also shown for each year. $R$ values correspond to the correlation between detrended OPA-nex results and estimates from available observations over the period. Grey shaded areas correspond to the standard deviation of the satellite-derived retrievals for each winter season.

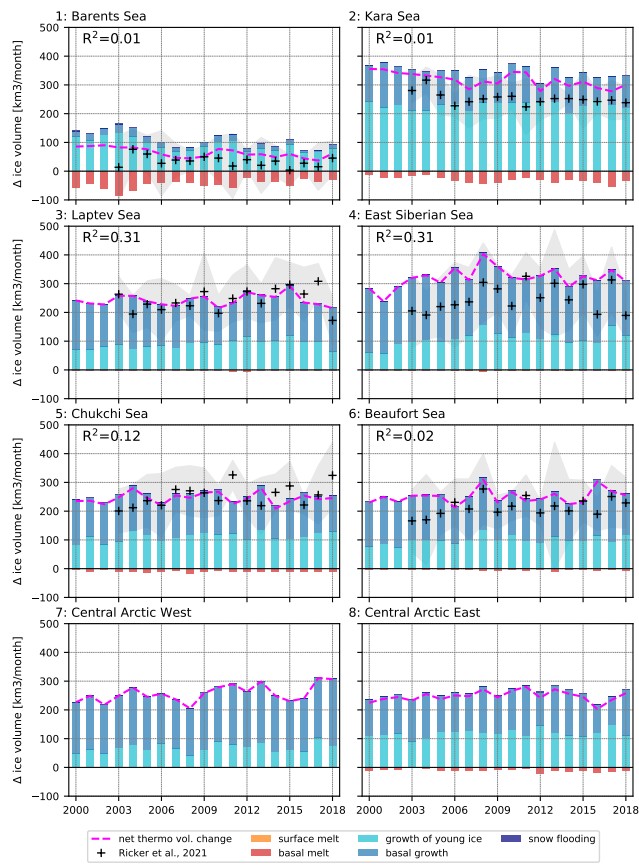

**Figure 8.** Temporal evolution of the sea ice mass balance due to thermodynamic processes for the winter months (November to March). Results are presented region per region in OPA-nex and compared to Ricker et al. (2021) estimates. The net contribution of thermodynamics is represented by the dashed magenta line. Contributions from the different growth and melt processes for each year are also shown. Surface melt is not visible here as thickness data are only available for the freezing season. $R^2$ values correspond to the correlation between detrended model results and estimates from observations over the period of availability. Grey shaded areas correspond to the standard deviation of the satellite-derived retrievals for each winter season.

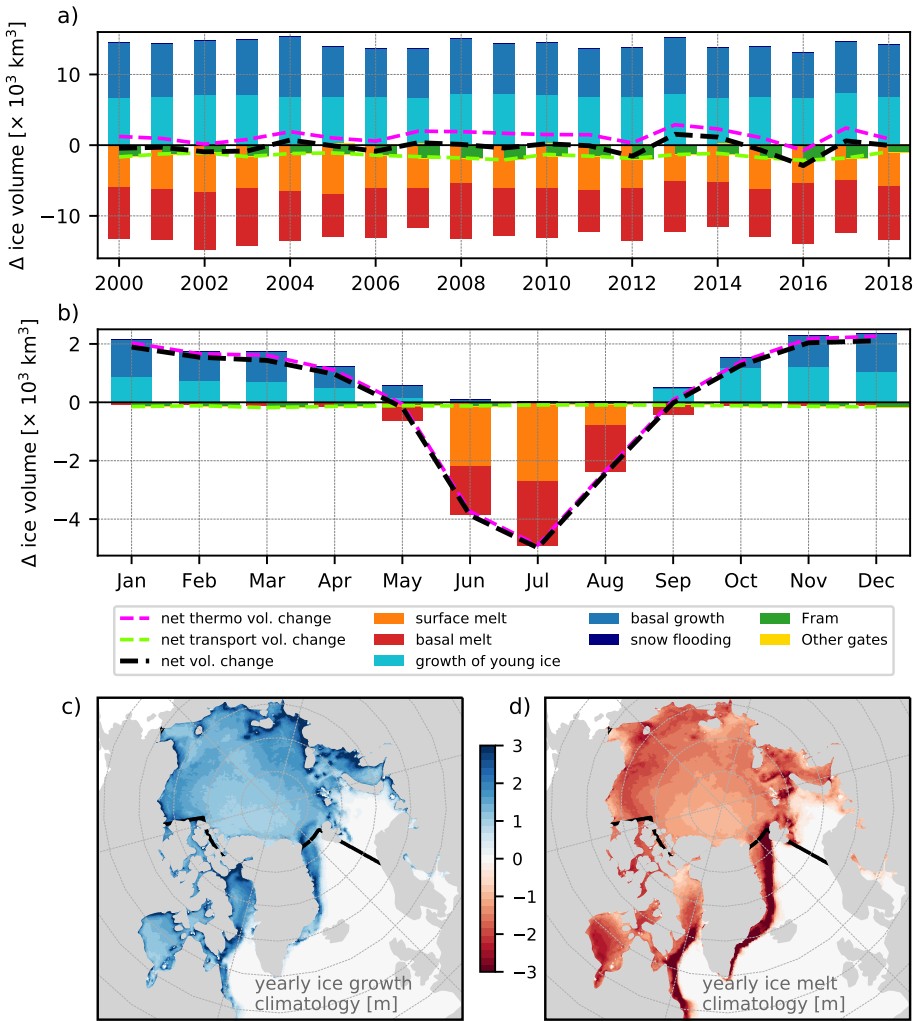

**Figure 9.** (a) Yearly evolution of the ice mass balance in OPA-nex distinguishing the different processes (thermodynamic and dynamic) contributing to ice volume gain/loss in the studied domain. (b) Monthly climatology of the ice mass balance over 2000–2018. Ice growth and melt distribution climatologies for these same periods are presented in panels (c) and (d) respectively.

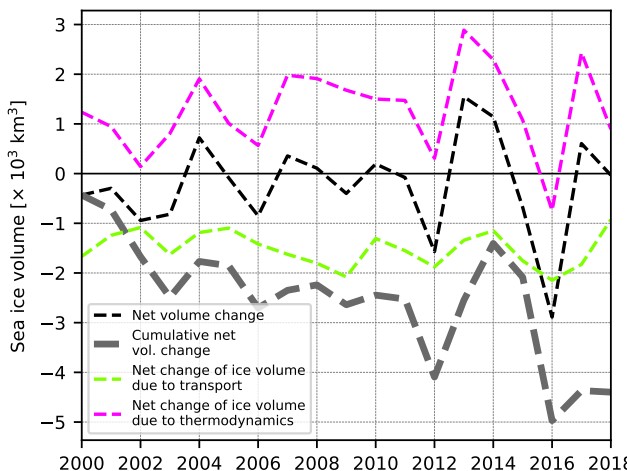

**Figure 10.** Temporal evolution of the yearly net sea ice volume change (dashed black line) and the cumulative yearly volume change (thick dashed grey line) over the period 2000–2018 within the study domain. The magenta and green lines represent the annual net volume change due to thermodynamic processes (magenta) and due to sea ice transport in/out of the domain (green line), as in Figure 9a.

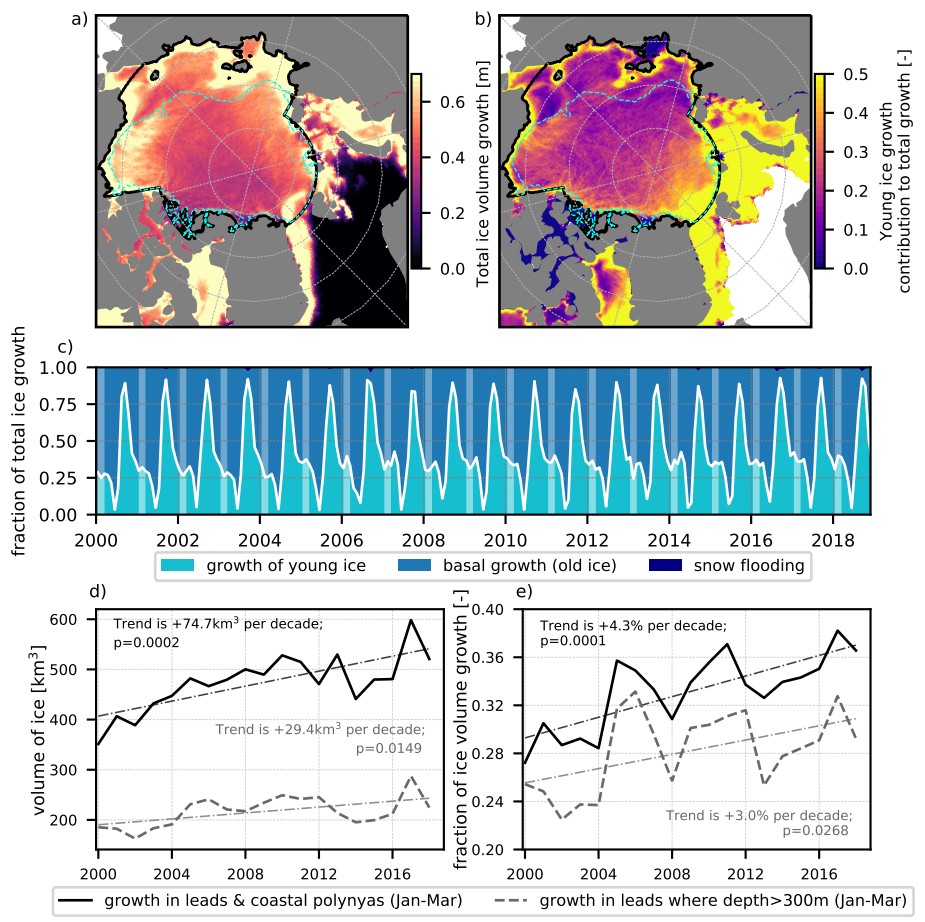

**Figure 11.** 2000–2018 climatology of the spatial distribution of ice volume (per area) growth in leads and polynyas in winter (a), and its contribution (ratio) to the total ice growth in winter (b). Panel (c) shows the evolution of this contribution compared to other ice growth processes over the period 2000–2018 within the domain delimited by the thick black solid line in panels (a,b). The temporal evolution of the integrated winter young ice volume growth and its contribution to total ice growth integrated over the domain are shown in panels (d,e) respectively (black solid lines). The grey dashed lines represent the same quantities, but for a sub-domain from which the regions shallower than 300m are excluded (cyan contour in panels a,b). We also display the trends associated with each line and their associated p-value.

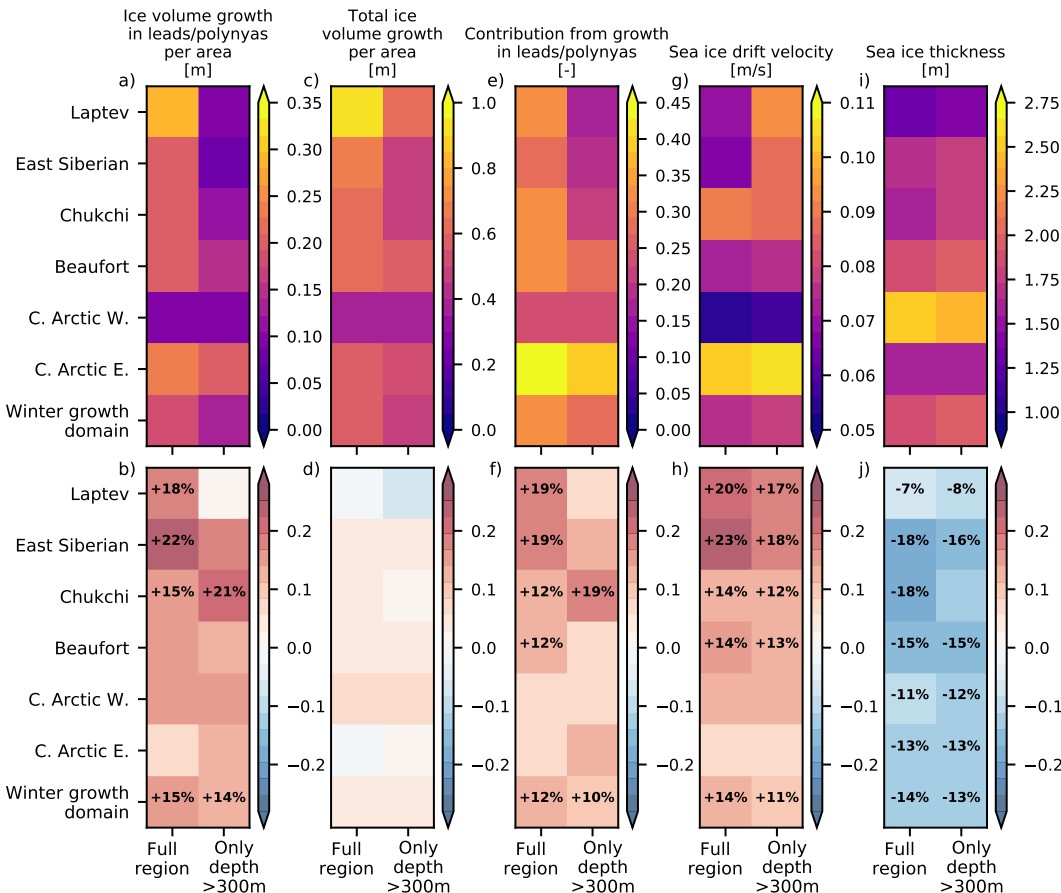

**Figure 12.** Top: Average quantities over the different sub-domains for, from left to right, winter ice volume growth in leads and polynyas, total winter ice growth, contribution of ice volume growth in leads and polynyas to total winter ice growth, sea ice drift velocity and thickness. Bottom: Trends over 2000–2018 in % (obtained by dividing the trend of each quantities by the mean values shown above) associated with each quantity. For each graph, the left column corresponds to the full sub-domains, and the right one only includes regions deeper than 300m.

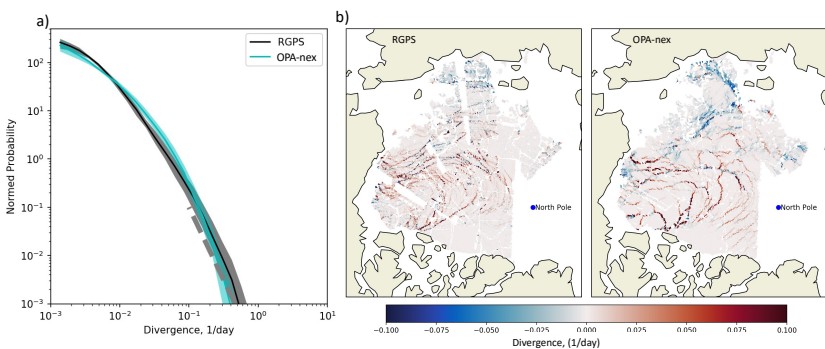

**Figure A1.** a) Probability density function of the divergent component of sea ice deformation rates computed from all OPA-nex snapshots in 2007 (between January 1st and April 30th) matching RGPS snapshots. The deformation snapshots are calculated over a timescale of 3 days. More details about these snapshots and the computation can be found in Ólason et al. (2022), section 3.1. The shaded area represents the standard deviation of the monthly variability of each probability density function. The dashed line is shown for reference and corresponds to a power law with an exponent equal to -3. (b) Maps of sea ice divergence (day$^{-1}$) for 17 February 2007 as observed by RGPS (right) and simulated in OPA-nex (left).