# Peer review of "Arctic sea ice mass balance in a new coupled ice-ocean model using a brittle rheology framework"

_The Cryosphere, 2022_

## Referee Comment (RC2)

Review of:

"*Arctic sea ice mass balance in a new coupled ice-ocean model using a brittle rheology framework*" by Guillaume Boutin, Einar Ólason, Pierre Rampal, Heather Regan, Camille Lique, Claude Talandier, Laurent Brodeau and Robert Ricker.

This manuscript presents a new coupled ice-ocean model (with neXtSIM for sea ice, OPA for the ocean) and discuss its performance in representing the Arctic sea ice mass balance, based on an 18 years long simulation (2000-2018). They describe their methods for coupling neXtSIM, a Lagrangian model, with OPA, an Eulerian model. This is done by first interpolating the neXtSIM fields onto an Eulerian mesh, such that the interpolated fields are used for the coupling. The study provides a detailed analysis of the modelled ice mass balance in terms of trends, inter-annual variability and seasonal cycles, and investigates both the thermodynamics and dynamical contributions to the mass balance. They show that the ice-ocean model captures the amount (25-35%) of ice growth occurring in leads and polynya, and that this portion has a positive trend mostly attributed to the coastal polynyas.

The manuscript is very clearly written, well detailed, and presents figures that are appropriate for the analysis. I find this manuscript very well prepared, and that the science (results and discussions) is of high quality. In all, this makes for a very good presentation of the new ice-ocean model, combined with an interesting study on the ice mass balance that will benefit the sea ice community.

I nonetheless have two points that I believe need to be address. First, the manuscript suffers from a couple of subjective statements about the included rheology, which do not relate to the provided analysis. While these statements are few and only found in the abstract, introduction and conclusion, they effectively leaves a first (and last) impression that the authors are pushing their rheology forward. In the context of a scientific manuscript, such subjective statements have a history of distracting readers from the actual analysis and to raise doubt on the transparency. I think it imperative that these statements, listed below, are rephrased or removed. Second, I also believe that more information could be given on the coupling, more particularly about the tuning of the ice drift, thickness and deformations, given that presenting the ice-ocean model is one of the main objectives of the manuscript.

For these reasons, I recommend this manuscript to be accepted for publication, after major revisions.

Major points:

- L7-8: *"Using this rheology enables the reproduction of the observed characteristics and complexity of fine-scale sea ice deformations with little dependency on the mesh resolution."* This implies that one needs the BBM rheology to have performance, which is far from being established. This performance may very well be related to the Lagrangian scheme. The dependency on the mesh resolution is intrinsic to all continuum

models, not to a given rheology. This is simply resolved by using a more appropriate and objective turn of phrase, such as "This rheology has been shown to reproduce…".

- L12: "*The model performs well*": unless accompanied by some quantifications, this remains vague and subjective.

- L14: "*Benefitting from the model's ability to reproduce fine-scale sea ice deformations, we estimate that the formation of sea ice in leads and polynyas contributes to 25%–35% of the total ice growth […]*": This statement made me expect some sort of demonstration of that benefit, but in the analysis, this benefit is assumed but not investigated. This is not that trivial to me, as we do not need fine-scale deformations to have growth and divergence within the pack-ice. Unless this benefit is shown, this should be rephrased.

- L99-100: This tuning is interesting but it is unclear what has actually been done. As this manuscript is presenting the coupled ice-ocean framework, I feel that this needs to be better described. In particular, it would be nice to have a figure that shows how the stress is chosen, and this tuning balance between drift and thickness distribution, and the deformation statistics. This is especially important as these are important parameters for the ice mass balance.

- L395: "*the inability of many models to correctly simulate LKFs*". This is a bit misleading and should be rephrased. The conclusion of SIREx is actually that all rheologies are able to produce LKFs, but none do so correctly due to a tendency to under-represent them.

- Last paragraph (L453-460) : "*Our results illustrate the interest of using a brittle rheology framework in ice–ocean coupled modelling* […]". This last paragraph is very subjective and brings conclusions that are by no means discussed in the analysis. Was is shown is that the new ice-ocean model is performing well. Attributing this to the rheology is, to me, not only reductive but inaccurate, as we are discussing a fully coupled ice-ocean model here. The extent at which the portion ice formation associated with pack ice divergence is dependent on the stated heterogeneity is also not demonstrated, and similar results could very well be obtained with other rheologies. I think the authors should focus on contributions demonstrated in the manuscript, which I believe are many and interesting by themselves.

Minor comments:

L5: (<5km) would be more accurate (see Hutter et al. 2022). Same in L30.

L32-34 : "*LKFs are related to the mechanical behaviour of the sea ice, and their absence in models…*". Too strong: they are not "absent" but under-represented.

L46-52 : This paragraph should be re-worked, I am not quite getting this modifications to the stress state. Is "stress state" used here as a synonym to rheology?

L56: Has this been portion been reported in classical models? If so, this could provide a measure on how much this 30% is being reproduced by (E)VP models, and would perhaps indicates the benefit of representing finer scale deformations.

L90-92: I believe that we could have a bit more information on the thermodynamics, as it is, after all, a significant contributor to the ice mass balance. For instance, is there a melt-pond scheme? How much do we expect results to be affected by the use of more sophisticated thermodynamics (i.e. including brine processes, snow model, etcs)?

L106: I believe that the BBM model has 2 time steps (dynamical and advection). I assume that the 450s time step for the ice model refers to the advection time step? Otherwise, this would mean that the model depends on an elastic component that is largely un-resolved. This needs to be clarified.

L113-120: What about the Lagrangian regridding?

L127: Here it is OPA-neX, but later it is OPE-nex.

L145-150: My understanding is that PIOMAS remains somewhat dependent on model outputs, and not without bias. A work or two on this would be useful.

L215: Not sure why "remarkable" is used. This needlessly adds subjectivity, unless the reason why this was unexpected is specified objectively.

L233-243: This is interesting. May this be related to the representation of old ice? This could explain this 2008 mark for the model performance, given the loss of old ice in observations after the 2007 summer.

L245: "*We first investigate*" I found "first" odd here

L255-156: "*This is consistent with the behaviour of PIOMAS*". This could be clarified. The same underestimation is seen in PIOMAS? What does it mean?

L262-265: I personally don't think that this paragraph is necessary.

L280: Is this similar to previous reports?

L285-291: This is a bit confusing to me. I understand in principle that anomalous melt in spring makes for anomalous surface to grow in Fall, but at the end you seem to say that anomalous growth in the fall also makes for anomalous melt in spring… This is going in circle to me, and if constant throughout the period, how do you get anomalies?

L294: Missing a reference for this dominance of basal melt south of Fram.

L300-301: This may be related to the mass-conserving snow-ice formation scheme. We find that this largely underestimates the snow-ice volume. In Turner et al. (2015) (ref below), the changes in the snow-ice parameterization was the largest contributor to pan-Arctic thickness changes associated with the implementation of the mushy layer in CICE.

L308: This is interesting and a comment could be added about what this implies. E.g, we know that the export of ice is has quite a variability associated with the AO. What this seem to suggest, is that the larger export is compensated with larger divergence and enhanced ice production?

L317: I find unclear what is meant as heterogeneity here. Is it used as a synonym to "leads"? We see the ice formation in leads and its contribution, but how is this a measure of heterogeneity?

Figure 8: I would specify right at the beginning that this only covers the winter, as the lack of melt is puzzling at first glance.

References: There are some errors in the references. For instance, Mehlmann et al., 2021 is incomplete. Some have errors in the DOIs. (E.g., Semtner 1976, Winton 2000, Zhang et al. 2003)

Mathieu Plante

Refs:
Turner, A. K., Hunke, E. C., & Bitz, C. M. (2013). Two modes of sea-ice gravity drainage: A parameterization for large-scale modeling. Journal of Geophysical Research: Oceans, 118 (5), 2279-2294. Retrieved from https://agupub.onlinelibrary.wiley.com/doi/abs/10.1002/jgrc.20171 doi: https://doi.org/10.1002/jgrc.20171

---

## Author Comment (AC1)

We thank the two referees for their very constructive comments and their suggestions which helped us improve the manuscript. We have tried to address these comments following their suggestions as explained in our answer below. We also made a small correction to Figure 4 as the time series was cut at the end of 2016 instead of 2018 in the original submission. In the following answer, PXLY refers to Page X Line Y of the updated version of the manuscript that is attached to our answer. References to manuscripts that were not in the original submission can be found at the end of our answer. We also attach a pdf with highlighted differences between the original submission and the updated manuscript.

Referee 1:

**Review of "Arctic sea ice mass balance in a new coupled ice-ocean model using a brittle rheology framework" by Boutin et al. (tc-2022-142)**

**The manuscript describes the first multi-decadal simulation of a coupled sea ice-ocean model with neXtSIM as the sea ice component. As such it is the first coupled ice-ocean simulation with a brittle rheology. The model system is driven by re-analysis (ERA5) and very well tuned to observed large scale features (ice extent, volume, mean drift speed). The manuscript then describes a mass (volume) balance of Arctic sea in great detail. As a main result, new sea ice formation contributes 25-35% of the total ice growth in winter. This contribution grows over time, mostly consistent with many previous studies.**

**The manuscript is generally well-written (see a few suggestions below) and has an easy to follow structure. The analysis appears to be very thorough and great care has been taken to map model results to observations (and not vice versa, as is often done). The results are interesting and warrant publication in TC.**

**There is one aspect that I find inconclusive and not supported by the presented results: The authors associate the new ice growth to the brittle sea ice dynamics that set neXtSIM apart from any other large-scale sea ice model. While I have no doubt that most of the ice growth takes place in open water or over thin ice (also in this model), I cannot see how the heterogeneity (at the grid scale) of the ice cover that this model features is an essential ingredient to the analysis. For a balanced analysis the authors need to "couch" their work differently.**

**Any sea ice model that I am aware of uses the sub grid scale parameterisation of ice concentration to simulate unresolved leads. With a simple diagnostic that records new ice formation over open water or the thinnest ice class (or great ice), it would be possible to repeat the present analysis with a sea ice model without brittle rheology and I would not expect very different results (although I may be wrong).**

**To show that the heterogeneity of the ice cover is an essential ingredient that we need to get right, there needs to be a comparison of a model with heterogeneity and without (not clear to me how that can be achieved cleanly, maybe with extra averaging The analysis of a single model simulation will only show that there is more ice production in areas of little or thin ice (which could be done with any model. In my opinion, the authors need to rephrase the corresponding parts of their manuscript or present clear evidence that supports their claims of this aspect.**

The way we understand the referee's comment, there are 2 aspects of our manuscript that are problematic:

- The link between young ice growth and heterogeneity is unclear in our analysis

- Our phrasing makes it sounds like other coarse resolution (>5km) models are not able to represent ice growth in leads (which is in line with the comments from referee 2), therefore implying that the model we use is "better" without making any comparison (which would not be straightforward, as noted by the referee).

We have addressed these two problems following the referee's specific comments. In general, we have tried to clarify our analysis and remove any sentence that could be interpreted as a demonstration of the benefits of using one model rather than another.

**More specifically:**

**page 1**

**l3: (second sentence of abstract) "These exchanges strongly depend on openings in the sea ice cover, which are associated with fine-scale sea ice deformations, but the importance of these processes remains poorly understood as most numerical models struggle to represent these deformations without using very costly horizontal resolutions". This is a strong claim that is unsupported, because even coarse models have sub grid parameterisations (sea ice concentration < 100%) that allow finite exchange. I have not seen any evidence that on average (10-100km to basin scale) the effects of fine-scale sea deformation are important for, e.g. heat exchange in coupled models. For ocean models it is very unclear (I am not aware of any work in that direction, please prove me wrong); regional atmospheric models have been used to illustrate the effects of leads on vertical and horizontal mixing, but in coupled simulations, atmosphere models are too coarse to resolve the forcing by leads. If the authors are aware of evidence that supports their claim, it needs to cited here (or in the introduction).**

We agree with the reviewer that these impacts have not been proven to be important (and have not been tested much in general) in coupled models, even though there have been suggestions they may matter (which we support citing Lüpkes et al., 2008; Marcq and Weiss, 2012; Steiner et al., 2013 in the introduction). We have therefore rephrased the beginning of this sentence as:

*P1L3: These exchanges have been suggested to strongly depend on…*

We also clarified this point and our motivation (to be able to eventually test these claims using models that can resolve these features) in the introduction:

*P2L26: These ubiquitous features, particularly leads, are expected to have a strong impact on ocean-ice-atmosphere interactions in polar regions (Lüpkes et al., 2008; Marcq and Weiss, 2012; Steiner et al., 2013), even though the importance of this impact remains unclear. To assess whether this impact is significant or not, numerical models need to represent the heterogeneity associated with LKFs in the ice cover, and therefore ensure a correct simulation of small-scale ice dynamics.*

**page 11**

**l317: "The impact of heterogeneity of the sea ice cover on winter ice production is visible in Figure 11a, and is clearly linked to the growth of young ice (Figure 11b). …" (maybe it is a good idea to have different color scales for 11a and b to stress that one is growth and the other a fraction of total growth).**

We changed the colormap of Figure 11a to clarify the difference between the two panels.

**While I do not question the heterogeneity of the ice cover, it does not become clear from this analysis that the "openings" are important for the net volume changes (growth) in the model. It's clear that ice can only grow over open water or thin ice. Sea ice models account for this by having at least 2 ice classes (thick ice and "thin ice including open water", see Hibler 79), most models have even more (e.g. 3 in neXtSIM or many more in CICE). This is the coarse resolution model's parameterisation of leads. Heat fluxes and growth rates are computed separately for the individual classes. For this to work, the ice distribution does not need to be heterogenous and it doesn't matter if the patterns "look similar to maps of observed ice divergence …" or not.**

We are not sure we understand the link between the point made by the referee here and the sentence that is pointed out. We agree that it would be possible to reproduce our analysis with another model, independently from the rheology. The most sensitive point seems to be that our phrasing implies that our model can estimate the right amount of ice grown in leads, while other (coarse resolution) models could not, making our analysis unbalanced. We agree with the referee that such a claim would be very unfair and that subgrid parameterizations in coarse models may be able to estimate the right amount of growth related to leads. We made this point clearer in the introduction of section 5.2.

Now, in terms of the importance of openings for volume changes, our analysis shows that, in our model, a significant part of the growth takes place in the openings, hence is associated with ice divergence. This is because sea ice divergence (hence openings in the ice) is the only way to create open water in winter in pack ice, and because young ice can only form in open water in the model. There is no other source term for young ice in the model. As a result, all the growth of this young ice results from sea ice divergence. Given the importance of young ice growth relative to total growth in pack ice in the model, our claim that openings have an important impact on the net volume growth in our simulation seems justified. The originality of our analysis is that we know that the model we use has a good representation of sea ice divergence, and is able to resolve divergent features like leads, even at 12~km resolution. The support for this claim can be found Ólason et al., 2022 and the newly added appendix A). A similar analysis could be done with a high-resolution sea ice model using a different rheology with the same argument.

We clarified the justification for our assumption in the paragraph, hoping it makes the link between "opening" and "young ice growth" clearer. As a result, the paragraph introducing section 5.2 has been rephrased as follows:

P11L335: *We now estimate the contribution of leads and polynyas to the winter ice mass balance. This estimate is based on the simulated ice formation in open water and ice growth in the young-ice category (see section 2.1). In winter and in pack ice, such ice growth will only take place where the ice has been recently diverging, because young ice quickly grows thick enough to be transferred to the "old ice" category (a few days at most). In the absence of divergence, the domain would be fully covered by old ice. The following analysis could be carried out with any sea-ice model with multiple ice thickness categories. However, the amount of ice produced in openings (i.e. leads and polynyas) in pack ice and its localisation are very likely to be strongly impacted by the ability of the model to reproduce the small-scale sea-ice dynamics. This is because the highest values of divergence rates (and deformation rates in general) in Arctic pack ice are very localised (Figure A1a,b), which would not be the case if the ice cover was homogeneous (e.g., Stern and Lindsay, 2009). For instance, Bouillon and Rampal (2015) found that in neXtSIM at 10 km resolution, 50% of the divergence in the Central Arctic was associated with only 5-10% of the surface area in the domain used for the analysis (this surface ratio would be 50% in the case of a homogeneous ice cover). Divergent ice motion, therefore, results primarily in the formation of localised leads in the central pack or of polynyas near the coast. The advantage of using neXtSIM in our analysis is that its ability to reproduce small-scale sea ice dynamics has been thoroughly evaluated before (see Ólason*

*et al., 2022, and appendix A). In addition, it has been shown that the model is able to capture rates of divergence consistent with observations, and relevant statistics of the observed lead fraction in the Central Arctic at spatial resolutions similar to the one used here (Ólason et al., 2021, 2022, and Figure A1).*

**page 14**

**l444: "The ability of the sea ice model to simulate fracturing and the subsequent sea ice deformations is used to assess the contributions of leads and polynyas to the mass balance." This claim is not supported by the presented work. It does not become clear that the fracturing and deformation as simulated by the brittle rheology affect the contribution of leads and polynyas to the mass balance.**

We agree that this statement would require further analysis to be fully supported. We rephrased this as:

P15L478: *We estimate the contribution of leads and polynyas to the winter mass balance. This contribution adds up to…*

**page 15**

**l453: "Our results illustrate the interest of using a brittle rheology framework in ice–ocean coupled modelling. This framework is able to capture the spatial and temporal heterogeneity of the ice cover, opening up the possibility to assess how this heterogeneity affects the ocean surface properties."**

**I think that this again is overselling neXtSIM's rheology. Heterogeneity at the grid scale should not be confused with realism. Further, grid point models always need a few grid points (order 5-10) to represent a feature. Very localised forcing at the grid scale may lead to local effects on the vertical mixing, which will then immediately be smoothed by horizontal processes. It is not clear if heterogeneous heat fluxes have a significant impact on mixed layer properties relative to smoothed heat fluxes. If there is evidence from the literature, please cite it.**

**Processes that involve thresholds, like biogeochemistry with minimal light requirements, the effect of heterogenous light conditions on net production, etc. are more plausible. Again, I have seen this claim a lot, without any proof or evidence from numerical modeling. Please cite the relevant literature.**

We removed this last paragraph as its subjectivity and some unsupported claims have been pointed out by the two referees. As a result, the study now ends with a short discussion from the previous paragraph on how to strengthen confidence in our results using available observations. We think it brings more focus to our contribution with no extrapolations on the advantages of such or such rheology, as recommended by the two referees (P16L481).

**L457-460: At higher horizontal resolution (which the authors have deemed too expensive earlier), non-brittle (VP) models also exhibit the heterogeneity (as cited in the introduction), so the advantage of neXtSIM does not become clear.**

This is true—this sentence has been removed from the rest of the paragraph.

**Data availability: All external (and open) data sources used in the study are listed, but availability of simulation data or code of this study is unclear.**

Monthly outputs of all quantities discussed in the manuscript are now available on zenodo as netcdf files. We also share the data used for each figure, also as netcdf. https://zenodo.org/record/7277523#.Y2UPIYLMIQM

The neXtSIM code is still in development and will be made open source in the coming months (in a dedicated publication).

**Minor comments and suggestions:**

**page 4**

**l99: "The stress values are chosen to match the observed large scale drift and thickness distribution as well as possible, while still maintaining good deformation patterns and statistics." It is not clear how this is done, what are "good deformation patterns and statistics"?**

These are the same metrics as in Olason et al. (2022), i.e. statistics such as the probability density function of sea ice deformations, and the qualitative assessment of the aspect of deformation features by comparing snapshots of deformations from RGPS observations and from the model. Following a recommendation from referee #2, we have added an appendix (P16L495) about the tuning of the model, in which we have added a Figure illustrating these diagnostics and supporting that the model's capability to reproduce small-scale dynamics is in line with Ólason et al. 2022, which was already an improvement over the neXtSIM version presented in Hutter et al. (2022) and Bouchat et al. (2022).

**l106: I stumbled over "twice the ice model time step", because a model timestep of 450 seconds would be short for VP model, but in the light of Plante et al 2020 or Dansereau et al 2016 (the only other ice models with brittle rheology that I am aware of), who both use timesteps of order 2 seconds to (marginally) resolve fast elastic waves, this seems like a very long time step for a elastic model. It would be useful to state here that the dynamics are solved with a much shorter time step (6s according to table1) to avoid confusion.**

This is a good point. We have added this precision in the introduction of the previous paragraph that details the neXtSIM setup used in this study.

P4L100: *The main (advection) model time step is 450s, with 120 sub-cycles used to solve the dynamics resulting in a dynamical time step of 6s.*

**page 5**

**L127 "OPA-neX" later "OPA-nex" is used (who a lower case "x").**

Fixed

**page 7**

**l186: "the internal stress is an important term in the Arctic mass balance", that's technically not correct. It may have an important effect on the mass balance, but it is a term in the momentum balance. Please rephrase.**

We rephrased it as:

P7L196: [...], *the internal stress is an important term in the momentum equation with the potential to affect the Arctic mass balance…*

**L188: "adding a degree of freedom to the simulation"? Unclear, what this is supposed to mean. I would remove it.**

We agree. We have removed this expression.

**l189: "careful", wording: I hope that everything reported here is based on "careful" analyses, so I would be careful with this adjective. I would replace it by something more descriptive, like "detailed", "thorough", if you really need to stress that you are "careful".**

We used "thorough" instead.

**l193: "The evaluation of …" Could be much shorter, e.g.**

**"The evaluation of small-scale dynamics of sea ice in the coupled neXtSIM/OPA setup provided no qualitative differences in sea ice deformations compared to a standalone setup (Olason et al. 2021a)."**

This is indeed much better, thank you very much for the suggestion.

**l196-198, Fig3: no numbers? Mean difference? RMSD?**

For Figure 3 we chose to give the IIEE, that is well suited to assess how a model reproduces the sea ice extent (more than the mean difference that may be low due to compensating errors). However, we agree we should discuss the IIEE before commenting on the model skills, the statements made L196-198 seem very subjective otherwise. We rephrased the beginning of the paragraph and put our conclusion at the end:

P7L206: *We start our evaluation with the sea ice extent (Figure 3). To quantify the agreement between OPA-nex and the OSI-SAF data over the study domain, we compute the integrated ice-edge error (IIEE), a metric used…*

**l198: "look at", colloquial, rephrase**

We replaced it with "compute", which is more appropriate in this case.

**Figure 4a and associated text (l206-214): Interannual variability is different from PIOMAS, notably the extreme minima in 2007, less so in 2012, are underestimated (not low enough). Instead, the OPA-nex timeseries tends to be more stable than the PIOMAS time series (less mean volume decrease and lower inter annual variability). Is this the ERA5 forcing or some model specifics/parameters?**

It is difficult to attribute these differences. It could be due to ERA5, to the differences in the thermodynamics or in the rheology, or to the data assimilation used in PIOMAS and not in our simulation. We added PIOMAS as a reference, as it is often used, and because it covers the whole period we study. However, in the manuscript, we prefer to investigate in more detail the differences between our model and observations than the differences with other model reanalysis.

We added a comment on that in the text:

P8L220: *A lot of factors could explain the discrepancies between OPA-nex and PIOMAS (differences in atmospheric forcings, in the dynamics and thermodynamics of the models, and the use of data assimilation in PIOMAS), and it is difficult to attribute these differences to one or the other of these factors.*

**page 8**

**l215: "remarkable": wording. I find this adjective not appropriate. This scientific MS should not be not about selling the results, and as sea ice drift is mostly determined by wind forcing, the agreement may not be as "remarkable" as claimed.**

We agree, the sentence has been rephrased.

P8L228: *The simulated drift generally shows a good agreement with the OSI-SAF data (Figure 5), with a low negative bias (-0.35 km/day on average from 2010 to 2018) and a low RMSE (3.82 km/day) for the freezing season (October to April), when most of the data are available.*

**L219 "overestimated", what is the reason for this overestimation when it did not happen before. Does the ice state change so that the ice becomes more mobile?**

In summer, the importance of the internal stress drops, independently from the rheology used in the model. This is because melt reduces sea ice concentration, which in turn reduces the ice strength, hence the internal stress. As a result, summer ice is almost in free drift, and the drift speed is largely controlled by the ratio between the ice-ocean and ice-atmosphere drag coefficients (see for instance Brunette et al., 2022, reference at the end of the document). Our comparison with OSI-SAF suggests this number could be re-tuned (as both drag coefficients are poorly constrained) in our simulation to get a better match with observations. However, as visible in Figure 5b, ice drift observations in summer are very uncertain, which makes us reluctant to re-tune the model in order to match exactly the summer drift. Details about the tuning, and the answer to the referee's question, are now given in Appendix A (added following referee's 2 request).

**l223: "component" here I would use "term", but that's a matter of taste**

We took the suggestion.

**L235: "agree very well" -> "agrees very well", although I would replace statements like these as much as possible by more quantitative statements ("very well" can mean anything).**

We rephrased it a bit and added the RMSE in the bracket (along with $R^2$) to support our statement:

P8L248: *OPA-nex captures this area flux very well (RMSE=20.28x10$^3$km$^2$/month ,$R^2$=0.81, [...]*

**page 9**

**l248: "estimations" -> estimates? (also elsewhere)**

Fixed

**l257: "significant"? Statistically significant? What is significant about this trend?**

Statistically significant, yes (i.e with a p-test result lower than 0.05). We added this precision in the text.

**l261: "is not well captured by OPA-nex", but for OPA-nex you can (and have done it) diagnose the individual terms of the model-thermodynamics, whereas Ricker et al could only indirectly estimate the thermodynamic growth. What happens if you use the same method as Ricker et al?**

We are not sure we understand this remark. Whether we use Ricker's indirect method (volume difference minus dynamic net change) or sum the growth terms minus the melt terms to retrieve the net growth, our results are unchanged (which is a good thing, otherwise it would mean our budget in each box is not closed). This has been done as a sanity check prior to 1st submission.

**l263: "look at", colloquial (also elsewhere), replace by "examine" or similar.**

Fixed ("analyse", "examine"...)

**l264: "This allows us to explore more deeply the links between the dynamic and thermodynamic contributions to the Arctic mass balance." How? Without explaining the "how", this sentence makes little sense and could be dropped.**

We agree. The "how" is explained in the next section, making this sentence not very useful and breaking the "flow" of the paper, so we dropped it as suggested.

**page 10**

**l304: Liu et al. (2020), please cite the numbers for the trends (or the range) for context.**

We have added more precision to this statement in general:

P11L326: *We find a statistically significant (p= 0.01) trend of -280 km3 per year over 2000–2008, which is within the range of sea ice volume trends (from both models and observations) discussed in Liu et al. (2020) (between −200 km₃ and −400 km3), but no significant trend for the period 2009–2018 (also as reported in Liu et al., 2020).*

**l313: "To do this, we assume that, …" this assumption is not specific to small scales or brittle rheology. This could be done for any model that has a sub grid scale ice concentration (i.e. virtually all sea ice models) at coarse resolution. (See main comment)**

This paragraph has been rewritten (see our answer to the main comment). We mention this point:

*P11L339: The following analysis could be carried out with any sea-ice model with multiple ice thickness categories.*

**page 12**

**l359: "look into more detail at", rephrase**

We replaced with "examine".

**page 13**

**L414/415 "was made of" -> consisted of?**

Fixed

**page 17**

**L531 Hutter et al has been pushlished in JGR: https://doi.org/10.1029/2021JC017666.**

The reference has been fixed.

**page 26**

**Fig7 caption: "estimations" -> estimates**

Fixed.

Referee 2:

**Review of:"Arctic sea ice mass balance in a new coupled ice-ocean model using a brittle rheology framework" by Guillaume Boutin, Einar Ólason, Pierre Rampal, Heather Regan, Camille Lique, Claude Talandier, Laurent Brodeau and Robert Ricker.**

**This manuscript presents a new coupled ice-ocean model (with neXtSIM for sea ice, OPA for the ocean) and discuss its performance in representing the Arctic sea ice mass balance, based on an 18 years long simulation (2000-2018). They describe their methods for coupling neXtSIM, a Lagrangian model, with OPA, an Eulerian model. This is done by first interpolating the neXtSIM fields onto an Eulerian mesh, such that the interpolated fields are used for the coupling. The study provides a detailed analysis of the modelled ice mass balance in terms of trends, inter-annual variability and seasonal cycles, and investigates both the thermodynamics and dynamical contributions to the mass balance. They show that the ice-ocean model captures the amount (25-35%) of ice growth occurring in leads and polynya, and that this portion has a positive trend mostly attributed to the coastal polynyas.**

**The manuscript is very clearly written, well-detailed, and presents figures that are appropriate for the analysis. I find this manuscript very well prepared, and that the science (results and discussions) is of high quality. In all, this makes for a very good presentation of the new ice-ocean model, combined with an interesting study on the ice mass balance that will benefit the sea ice community.**

**I nonetheless have two points that I believe need to be address. First, the manuscript suffers from a couple of subjective statements about the included rheology, which do not relate to the provided analysis. While these statements are few and only found in the abstract, introduction and conclusion, they effectively leaves a first (and last) impression that the authors are pushing their rheology forward. In the context of a scientific manuscript, such subjective statements have a history of distracting readers from the actual analysis and to raise doubt on the transparency. I think it imperative that these statements, listed below, are rephrased or removed. Second, I also believe that more information could be given on the coupling, more particularly about the tuning of the ice drift, thickness and deformations, given that presenting the ice-ocean model is one of the main objectives of the manuscript.**

**For these reasons, I recommend this manuscript to be accepted for publication, after major revisions.**

**Major points:**

**- L7-8: "Using this rheology enables the reproduction of the observed characteristics and complexity of fine-scale sea ice deformations with little dependency on the mesh resolution." This implies that one needs the BBM rheology to have performance, which is far from being established. This performance may very well be related to the Lagrangian scheme. The dependency on the mesh resolution is intrinsic to all continuum models, not to a given rheology. This is simply resolved by using a more appropriate and objective turn of phrase, such as "This rheology has been shown to reproduce...".**

We agree with the referee and we replaced our initial phrasing with their suggestion.

**- L12: "The model performs well": unless accompanied by some quantifications, this remains vague and subjective.**

This statement (and the rest of the sentence) has been rewritten to be more accurate:

*P1L12: Model values show a good match with observations, remaining within the estimated uncertainty, and the interannual variability of the dynamic contribution to the winter mass balance is generally well captured.*

**- L14: "Benefitting from the model's ability to reproduce fine-scale sea ice deformations, we estimate that the formation of sea ice in leads and polynyas contributes to 25%–35% of the total ice growth [...]": This statement made me expect some sort of demonstration of that benefit, but in the analysis, this benefit is assumed but not investigated. This is not that trivial to me, as we do not need fine-scale deformations to have growth and divergence within the pack-ice. Unless this benefit is shown, this should be rephrased.**

We agree that the benefit (or not) of having a good representation of fine-scale dynamics is not proven in this study, and that this sentence should be rephrased to be clearer about how to interpret it.

We suggest:

*P1L15: Using the ability of the model to represent divergence motions at different scales, we investigate the role of leads and polynyas in ice production. We suggest a way to estimate the contribution of leads and polynyas to ice growth in winter, and we estimate this contribution to add up to 25%--35% of the total ice growth in pack ice from January to March. This contribution shows a significant increase over 2000--2018.*

**- L99-100: This tuning is interesting but it is unclear what has actually been done. As this manuscript is presenting the coupled ice-ocean framework, I feel that this needs to be better described. In particular, it would be nice to have a figure that shows how the stress is chosen, and this tuning balance between drift and thickness distribution, and the deformation statistics. This is especially important as these are important parameters for the ice mass balance.**

We agree that this is interesting. A large part of this tuning and the effects of the different parameters are already presented in Ólason et al. 2022. We understand that the level of detail

given in the main text is too low to really inform the reader about what has been done, but we thought adding the necessary details to fully understand the tuning process would "break the flow" of the manuscript. To include this information, we suggest adding this information in an appendix, as we have done in the updated version of the manuscript (see P16L493, Appendix A).

**- L395: "the inability of many models to correctly simulate LKFs". This is a bit misleading and should be rephrased. The conclusion of SIREx is actually that all rheologies are able to produce LKFs, but none do so correctly due to a tendency to under-represent them.**

We agree and rephrased it as:

P14L429: *and the under-representation of LKFs by most models for spatial resolutions larger than ~5 km (Hutter et al., 2022; Bouchat et al., 2022).*

**- Last paragraph (L453-460) : "Our results illustrate the interest of using a brittle rheology framework in ice–ocean coupled modelling [...]". This last paragraph is very subjective and brings conclusions that are by no means discussed in the analysis. Was is shown is that the new ice-ocean model is performing well. Attributing this to the rheology is, to me, not only reductive but inaccurate, as we are discussing a fully coupled ice-ocean model here. The extent at which the portion ice formation associated with pack ice divergence is dependent on the stated heterogeneity is also not demonstrated, and similar results could very well be obtained with other rheologies. I think the authors should focus on contributions demonstrated in the manuscript, which I believe are many and interesting by themselves.**

We removed this last paragraph as its subjectivity and some unsupported claims have been pointed out by the two referees. As a result, the study now ends with a short discussion from the previous paragraph on how to strengthen confidence in our results using available observations. We think it brings more focus to our contribution with no extrapolations on the advantages of such or such rheology, as recommended by the two referees.

**Minor comments:**

**L5: (<5km) would be more accurate (see Hutter et al. 2022). Same in L30.**

We replaced 2km with 5km as suggested.

**L32-34 : "LKFs are related to the mechanical behaviour of the sea ice, and their absence in models...". Too strong: they are not "absent" but under-represented.**

We replaced absence with under-representation as suggested.

**L46-52 : This paragraph should be re-worked, I am not quite getting this modifications to the stress state. Is "stress state" used here as a synonym to rheology?**

We have rephrased the beginning of this paragraph to make it clearer. However, we are not sure we understand this comment completely as we did not use the expression "stress state" in the manuscript.

P2L49: *Choosing which rheology to use in a sea ice model is likely to have an impact on the modelled sea ice mass balance in the Arctic (Steele et al., 1997). One of the reasons is that the internal stress of the ice, the term related to the sea ice rheology in the momentum equation, impacts the net transport of ice between regions.*

**L56: Has this been portion been reported in classical models? If so, this could provide a measure on how much this 30% is being reproduced by (E)VP models, and would perhaps indicates the benefit of representing finer scale deformations.**

Not to our knowledge, at least not directly linking ice growth and divergence. We now mention in section 5.2 that our analysis could be done using any other model using 2 or more ice categories, independently of the rheology (P11L339, "*The following analysis could be carried out with any sea-ice model with multiple ice thickness categories*").

A fair comparison with "classical" models would also not be so straightforward as the diversity of parameterizations (for the dynamics, thermodynamics, ice thickness distribution) used by the community would make it hard to attribute the difference to such or such process only. In the absence of similar estimates using a VP rheology, we prefer to not comment about what the differences could be.

**L90-92: I believe that we could have a bit more information on the thermodynamics, as it is, after all, a significant contributor to the ice mass balance. For instance, is there a melt-pond scheme? How much do we expect results to be affected by the use of more sophisticated thermodynamics (i.e. including brine processes, snow model, etcs)?**

We agree some more information could be added, in particular about the albedo as it partly answers the referee's question about melt-ponds. We do not use a melt-pond scheme, but the albedo scheme used here accounts for the presence of melt ponds by decreasing the albedo values as sea ice temperature increases.

P4L94: *but the albedo scheme we use (the same as the standard albedo scheme "ccsm3" used in CICE, Hunke et al., 2017) accounts for the effect of melt ponds by reducing the albedo value when the surface temperature of sea ice increases.*

While we understand the interest of the referee's second comment, we are not in favour of commenting on the potential effects of adding such or such feature to the model thermodynamics, as it is very difficult to predict what to expect. It is likely some processes are missing (melt pond, snow transport by wind…) and are compensated by others; adding a process would likely require changes in the tuning parameter values to re-obtain a comparable quality of the results for the metrics we used in our evaluation (like the sea ice volume). We added a comment in the text:

P4L96: *It is likely that the use of an explicit melt-pond scheme (e.g. Flocco et al., 2010), or more complex representations of processes related to brine (Vancoppenolle et al., 2009) (instead of a constant salinity here) or snow would affect the sea ice extent and thickness in our results, but the effect of using another parameterization could only be assessed after a re-tuning of the model (as in Zampieri et al., 2021).*

**L106: I believe that the BBM model has 2 time steps (dynamical and advection). I assume that the 450s time step for the ice model refers to the advection time step? Otherwise, this would mean that the model depends on an elastic component that is largely un-resolved. This needs to be clarified.**

Yes, 450s is the advection time step, while the dynamical time step is 6s (120 subcycles per time step), which allows the model to resolve the elastic component.

We have added this precision in the introduction of the previous paragraph that details the neXtSIM setup used in this study.

P4L100: *The main (advection) model time step is 450s, with 120 sub-cycles used to solve the dynamics resulting in a dynamical time step of 6s.*

**L113-120: What about the Lagrangian regridding?**

The interpolation weights are recomputed after each Lagrangian regridding. We added this precision to the main text, P5L128.

**L127: Here it is OPA-neX, but later it is OPE-nex.**

Fixed.

**L145-150: My understanding is that PIOMAS remains somewhat dependent on model outputs, and not without bias. A work or two on this would be useful.**

This is true. We added a few words about that in this paragraph:

P6L155: *PIOMAS data are the results of coupled ocean–sea-ice model simulations with the daily assimilation of satellite sea ice concentration and sea surface temperature. The main interest of the PIOMAS dataset is that it is available for the whole simulated period and has been thoroughly evaluated against ice thickness observations (e.g. Schweiger et al., 2011; Laxon et al., 2013; Stroeve et al., 2014), meaning that some of its biases are known.*

And also when comparing our results with PIOMAS (following a comment of referee 1 who was asking about the reasons behind the differences between OPA-nex and PIOMAS ice volume):

P8L220: *A lot of factors could explain the discrepancies of between OPA-nex and PIOMAS (differences in atmospheric forcings, in the dynamics and thermodynamics of the models, and the use of data assimilation in PIOMAS), and it is difficult to attribute these differences to one or the other of these factors.*

**L215: Not sure why "remarkable" is used. This needlessly adds subjectivity, unless the reason why this was unexpected is specified objectively.**

We agree (and so does referee 1). We rephrased it as:

P8L228: *The simulated drift generally shows a good agreement with the OSI-SAF data (Figure 5), with a low negative bias (-0.35 km/day on average from 2010 to 2018) and a low RMSE (3.82 km/day) for the freezing season (October to April), when most of the data are available.*

**L233-243: This is interesting. May this be related to the representation of old ice? This could explain this 2008 mark for the model performance, given the loss of old ice in observations after the 2007 summer.**

It could be. We added a comment on that:

P9L256: *The period 2007--2008 corresponds to a large loss of old ice in the Arctic (Kwok, 2018), which suggests that this underestimate could be due to a negative bias in the thickness of the older ice prior to 2008 in the model.*

**L245: "We first investigate" I found "first" odd here.**

We rephrased it as: "We now investigate".

**L255-156: "This is consistent with the behaviour of PIOMAS". This could be clarified. The same underestimation is seen in PIOMAS? What does it mean?**

We rephrased to make our sentence clearer:

P9L271: *This overestimation of ice growth in these seas is also visible in the data from PIOMAS, ...*

**L262-265: I personally don't think that this paragraph is necessary.**

We removed half of it. We think what remains helps the reader to transition from section 4 to 5. But the last sentence was indeed unclear and not useful.

**L280: Is this similar to previous reports?**

In a sense, yes, as previous reports generally indicate a strong domination of Fram Strait for the overall sea ice export out of the Arctic basin (85% to 90% depending on the reference), but we could not find any comparison of the net import/export of sea ice through each of the gates of the Arctic Basin we use here. The most relevant reference we could find is Carmack et al. (2016) which compiled estimates from previous studies to do a freshwater budget of the Arctic. They estimate that the solid freshwater flux (=sea ice) out of Fram Strait is ~2000km3/year, while 300km3/year exit the Arctic through Davis Strait (which would correspond to our gates in the Canadian Arctic Archipelago and through Nares Strait). The main inflow of sea ice comes from Bering Strait and is estimated to be ~100km3/year. That means that, if we distinguish Fram Strait from the other gates, it should represent ~90% of the net total export (2000km3/2200km3) out of the Arctic. In our case, the transport through all gates other than Fram Strait almost cancels out over the study period. This is likely because i) our domain boundary is north of the Canadian Arctic Archipelago (CAA), and is not directly comparable to the boundary at Davis Strait since sea ice can recirculate in and out of our domain and, further, some of the ice export out of Davis Strait may be ice that formed locally in the CAA, and ii) we very likely underestimate the amount of ice exiting the domain through Nares Strait (that is almost 0 in the model, while observations suggest ~100km3/year). This is because a horizontal resolution of 12km is too coarse to simulate the sea ice flux (in and out) in this narrow strait. We have added some comments about this in the text.

P10L295: *Previous reports suggest that Fram Strait represents ~90% of the net sea ice export of the Arctic, the second main source of export being through Davis Strait, south of our domain (Carmack et al., 2016). In our case, the contributions from all gates other than Fram Strait almost cancel out. This is likely because i) our domain boundary is north of the Canadian Arctic Archipelago and therefore not directly comparable to Carmack et al., (2016) (for instance, we miss all the ice that forms there and is then exported through Davis Strait), and ii) 12km is too coarse to resolve the outflow through Nares Strait, leading to an underestimated export through this narrow gate (only 1km3/year in the model, while observations suggest an average up to ~190km3/year over 2017–2019, Moore et al., 2021)*

**L285-291: This is a bit confusing to me. I understand in principle that anomalous melt in spring makes for anomalous surface to grow in Fall, but at the end you seem to say that anomalous growth in the fall also makes for anomalous melt in spring... This is going in circle to me, and if constant throughout the period, how do you get anomalies?**

Having re-read the text, the reviewer makes a good point here. We removed the mention of a constant ratio between melt and freeze as it is indeed confusing. Instead, we comment on what controls the changes in sea ice volume in the domain we use which answers the question of how we get anomalies and makes a better transition to the next paragraph.

P10L310: *This is most likely because strong melt events lead to large areas of open water and thinner ice at the end of the summer, enhancing the refreezing in the next autumn and winter (Petty et al., 2018). We do not find any trend in sea ice growth nor melt using this domain, and large changes in the total ice volume (as in 2002, 2012, 2014 or 2016) are mostly associated with the interannual variability of the balance between melt and growth (Figure 10).*

**L294: Missing a reference for this dominance of basal melt south of Fram.**

We added two references to support this statement, one using model analysis (Bitz et al., 2005) and one using ice mass balance buoys (Lei et al., 2018).

P11L316: *while the basal melt dominates south of Fram Strait (outside the study domain), likely because sea ice encounters warmer surface waters in the Greenland Sea (Bitz et al., 2005 ; Lei et al., 2018).*

**L300-301: This may be related to the mass-conserving snow-ice formation scheme. We find that this largely underestimates the snow-ice volume. In Turner et al. (2015) (ref below), the changes in the snow-ice parameterization was the largest contributor to pan-Arctic thickness changes associated with the implementation of the mushy layer in CICE.**

This is a good point. We now mention this potential underestimation:

P11L323: *This contribution may, however, be underestimated by the mass-conserving snow-ice formation scheme used (Turner et al. 2015).*

**L308: This is interesting and a comment could be added about what this implies. E.g, we know that the export of ice is has quite a variability associated with the AO. What this seem to suggest, is that the larger export is compensated with larger divergence and enhanced ice production?**

It could well be, but we are reluctant to hypothesize on this as the variability of the export always remains very low compared to the one of ice production (which is what we state here) and the two do not seem to be correlated in general.

**L317: I find unclear what is meant as heterogeneity here. Is it used as a synonym to "leads"? We see the ice formation in leads and its contribution, but how is this a measure of heterogeneity?**

We agree this is unclear in this sentence and rephrased it as:

P12L351: "The impact of leads and polynyas…"

**Figure 8: I would specify right at the beginning that this only covers the winter, as the lack of melt is puzzling at first glance.**

We agree, it has been fixed.

**References: There are some errors in the references. For instance, Mehlmann et al., 2021 is incomplete. Some have errors in the DOIs. (E.g., Semtner 1976, Winton 2000, Zhang et al. 2003)**

We double-checked the references and corrected the ones that were incomplete. As noted by the referee, some DOIs include "surprising" characters in them, but we checked and it does not seem to be an error (for instance in Semtner 1976, Winton 2000, Zhang et al. 2003).

**New references :**

Brunette, C., Tremblay, L. B., and Newton, R.: A new state-dependent parameterization for the free drift of sea ice, The Cryosphere, 16, 533–557, https://doi.org/10.5194/tc-16-533-2022, 2022.

[revised manuscript text omitted]

---

## Referee Report (RR1)

2[nd] Review of:
"*Arctic sea ice mass balance in a new coupled ice-ocean model using a brittle rheology framework*" by Guillaume Boutin, Einar Ólason, Pierre Rampal, Heather Regan, Camille Lique, Claude Talandier, Laurent Brodeau and Robert Ricker.

This manuscript presents a new coupled ice-ocean model (with neXtSIM for sea ice, OPA for the ocean) and discuss its performance in representing the Arctic sea ice mass balance based on an 18 years long simulation (2000-2018). They describe their methods for coupling neXtSIM, a Lagrangian model, with OPA, an Eulerian model. They provide a detailed analysis of the modelled ice mass balance in terms of trends, inter-annual variability and seasonal cycles of each thermodynamics and dynamics contributors. They show that the ice-ocean model captures the amount (25-35%) of ice growth occurring in leads and polynya, and that this portion has a positive trend mostly attributed to the coastal polynyas.

I commend the work made by the authors to clearly and genuinely address all of my previous comments in the manuscript. All pointed statements have been either corrected or removed, and details about the model tuning has been added as an appendix. Altogether, this makes for a strong and interesting analysis that we interest the ice modeling community.

I therefore recommend its publication, after addressing a few remaining clarifications:

- Abstract L8 "*with little dependency on the mesh resolution*", as well as at P2, L38-39: "*regardless of the horizontal resolution used*". This should be rephrased: I believe that the authors here refer to the production of LKFs and heterogeneity at relatively coarse resolution, but not that the model is insensitive to the choice of resolution (which I doubt any model could claim to be). I suggest changing for (at both L8 and L38) "at relatively coarse resolutions".

- L26-28: This seems self-contradicting (they are expected to have a strong impact but their importance is unclear). To me, the impact of LKFs and heterogeneity in the real world is quite clear, but the necessity to resolve the very fine scale to represent this impact in a sea ice model is not. E.g., what is the benefit of resolving the smaller-scale heterogeneity, versus representing its impact via sub-grid parameterizations?

- L49: Steele et al. 1997: I am not sure if this reference still appropriate in this modified phrase. They look at the contribution of the internal stress terms in the sea-ice force balance, but not the ice mass balance nor do they discuss the choice of rheology.

- L340-349: This paragraph is more confusing than enlightening: it meanders around the unclear relation between deformation rates and ice growth in leads, while trying to present the small-scale dynamics as an advantage for the representing this ice growth. I believe that this should rather be brought as a question of interest, with neXtSIM as a new means to answer it. For instance, what do we expect, in terms of ice growth, from a localised lead vs. a non localised lead, especially considering the use of an ITD? What does it changes, in terms of ice growth, that a model has the right rates of divergence (or not)?

Minor typos:

- L76: Typo, remove "make"

- L101: Typo, add space between the point and the new phrase

- L125-128: This probably means that NEMO sees a smoothed version of neXtSIM?

- Table 1: 3$^{rd}$ row, I think the convention for units should be kPa m$^{-3/2}$

- Figure 5: units convention, I think it should be [km day$^{-1}$] instead of [km/day]

Congratulations to all the authors for this nice analysis,

Mathieu Plante

---

## Author Response (AR2)

**Clarifications:**

**- Abstract L8 "with little dependency on the mesh resolution", as well as at P2, L38-39: "regardless of the horizontal resolution used". This should be rephrased: I believe that the authors here refer to the production of LKFs and heterogeneity at relatively coarse resolution, but not that the model is insensitive to the choice of resolution (which I doubt any model could claim to be). I suggest changing for (at both L8 and L38) "at relatively coarse resolutions".**
This is a good point. We followed the referee's suggestion.

**- L26-28: This seems self-contradicting (they are expected to have a strong impact but their importance is unclear). To me, the impact of LKFs and heterogeneity in the real world is quite clear, but the necessity to resolve the very fine scale to represent this impact in a sea ice model is not. E.g., what is the benefit of resolving the smaller-scale heterogeneity, versus representing its impact via sub-grid parameterizations?**

We agree with the referee that this sentence is a bit self-contradicting. However, the question suggested in the comment ("what is the benefit of resolving the smaller-scale heterogeneity, versus representing its impact via sub-grid parameterizations?") is not exactly the one we address in the manuscript (and as pointed out by the other referee, this is not one we can address with this setup), it might therefore be misleading to mention it in our introduction. Instead, we suggest rephrasing this sentence to insist on the fact that we use our model to quantify the impact of LKFs at a Pan-Arctic scale.

*L26: These ubiquitous features, particularly leads, are expected to have a strong impact on ocean-ice-atmosphere interactions in polar regions (Lüpkes et al., 2008; Marcq and Weiss, 2012; Steiner et al., 2013), but this impact at a pan-Arctic scale has not yet been quantified.*

**- L49: Verify reference to Steele et al. 1997: I am not sure it is an accurate ref to use here.**

We agree that this reference is not accurate for this sentence (as the impact of rheology on mass balance is not discussed in their article), but it supports the following sentence (about the fact that internal stress matters for sea ice transport as it affects motion and thickness). We therefore moved this reference to the end of the next sentence.

**- L340-349: This paragraph is more confusing than enlightening: it meanders around the unclear relation between deformation rates and ice growth in leads, while trying to present the small-scale dynamics as an advantage for the representing this ice growth. I believe that this should rather be brought as a question of interest, with neXtSIM as a new means to answer it. For instance, what do we expect, in terms of ice growth, from a localised lead vs. a non localised lead, especially considering the use of an ITD? What does it change, in terms of ice growth, that a model has the right rates of divergence (or not)?**

We have rephrased to make more explicit the question of interest, but are reluctant to put too much emphasis on things (i.e. the impact of explicitly resolving leads or not) that are beyond the analysis presented in this paper, as it was one of the main negative comment of the other referee. We are also a bit confused by the point between divergence and the ITD.

To represent the effects of leads with an ITD in the absence of divergence, there is a need to constrain the ITD shape to ensure the presence of open water or very thin ice (which would be the effect of divergence). The simplest way would be to, for instance, cap the concentration of thick ice, so that each cell includes at least a small fraction of thin ice or open water to represent the leads (this is a possibility in the LIM3 model). Just "using" an ITD and letting it evolve prognostically is not sufficient. This may be what the referee means with the expression "non-localized leads" (as they are not related to any physical process, and therefore their spatial distribution is likely to be uniform). The paragraph has been rewritten with these comments:

L340: *This is because the highest values of divergence rates (and deformation rates in general) in Arctic pack ice are very localised (Figure A1a,b), which would not be the case if the ice cover was homogeneous (e.g., Stern and Lindsay, 2009). For instance, Bouillon and Rampal (2015) found that in neXtSIM at 10 km resolution, 50% of the divergence in the Central Arctic was associated with only 5-10% of the surface area in the domain used for the analysis (this surface ratio would be 50% in the case of a homogeneous ice cover). Divergent ice motion, therefore, results primarily in the formation of localised leads in the central pack or of polynyas near the coast. An underestimate of divergence rates, which "standard" sea ice models run at resolutions coarser than 5km tend to do (Hutter et al., 2021), would imply a subsequent underestimation of ice production in winter if there is not a sufficient parameterization to represent the effect of leads. This parameterization can be done using, for instance, a minimum value for the lead fraction in each grid cell, resulting in a more uniform distribution of lead growth over the domain (as this can be done in the LIM3 model, Rousset et al., 2015). The importance of resolving leads versus using parameterizations to represent the ice growth in leads in numerical models has not been assessed to our knowledge. This would likely require a model comparison between a model which captures divergence rates well and another one using a parameterization for leads, which is out of the scope of this study. Instead, we focus on estimating the importance of ice production in leads in our simulation, as this has not been estimated at a Pan-Arctic scale before. The advantage of using neXtSIM in our analysis is that its ability to reproduce small-scale sea ice dynamics has been thoroughly evaluated before (see Ólason et al., 2022, and appendix A). In addition, it has been shown that the model is able to capture rates of divergence consistent with observations and relevant statistics of the observed lead fraction in the Central Arctic at spatial resolutions like the one used here (Ólason et al., 2021, 2022, and Figure A1).*

**Minor typos:**

**- L76: Typo, remove "make"**
Done
**- L101: Typo, add space between the point and the new phrase**
Done
**- L125-128: This probably means that NEMO sees a smoothed version of neXtSIM?**
The triangles of the neXtSIM mesh are constrained to have side lengths within 10% of the side lengths of the exchange grid cells. Interpolation of the fields is inevitable since the two models are running on different grids, and the term "remapping" may therefore be more adequate than "smoothing". The interpolation approach also ensures that any smoothing is minimal

and very unlikely to impact the dynamics discussed in the paper. As interpolation is inevitable and smoothing minimal, we think that mentioning the word "smoothing" in the manuscript would be more confusing to the reader than the current way we describe the interpolation in L125-128.

**- Table 1: 3rd row, I think the convention for units should be kPa m-3/2**
Done
**- Figure 5: units convention, I think it should be [km day-1] instead of [km/day]**
Done